# USP15-dependent lysosomal pathway controls p53-R175H turnover in ovarian cancer cells

Achuth Padmanabhan[1,2,3], Nicholes Candelaria[1,2,3], Kwong-Kwok Wong [4], Bryan C. Nikolai[1,2,3], David M. Lonard[1,2,3], Bert W. O'Malley[1,2,3] & JoAnne S. Richards[1,2,3]

Gain-of-function p53 mutants such as p53-R175H form stable aggregates that accumulate in cells and play important roles in cancer progression. Selective degradation of gain-of-function p53 mutants has emerged as a highly attractive therapeutic strategy to target cancer cells harboring specific p53 mutations. We identified a small molecule called MCB-613 to cause rapid ubiquitination, nuclear export, and degradation of p53-R175H through a lysosome-mediated pathway, leading to catastrophic cancer cell death. In contrast to its effect on the p53-R175H mutant, MCB-613 causes slight stabilization of p53-WT and has weaker effects on other p53 gain-of-function mutants. Using state-of-the-art genetic and chemical approaches, we identified the deubiquitinase USP15 as the mediator of MCB-613's effect on p53-R175H, and established USP15 as a selective upstream regulator of p53-R175H in ovarian cancer cells. These results confirm that distinct pathways regulate the turnover of p53-WT and the different p53 mutants and open new opportunities to selectively target them.

[1] Department of Molecular and Cellular Biology, Baylor College of Medicine, Houston, TX 77030, USA. [2] Dan L. Duncan Cancer Center, Baylor College of Medicine, Houston, TX 77030, USA. [3] Center for Reproductive Medicine, Baylor College of Medicine, Houston, TX 77030, USA. [4] Department of Gynecologic Oncology and Reproductive Medicine - Research, Division of Surgery, The University of Texas MD Anderson Cancer Center, Houston, TX 77030, USA. Correspondence and requests for materials should be addressed to A.P. (email: achuth.padmanabhan@bcm.edu)

Tumor protein 53 (*p53*) is a transcription factor that regulates genes involved in cell cycle progression, DNA damage response, metabolism, and apoptosis[1–4]. Loss of p53 function is often associated with uncontrolled proliferation, increased genomic instability, and malignant transformation[5–8]. Mutations in *p53* are observed in over 50% of human malignancies, making it the most common genetic alteration in cancer[1,9]. Cancer genome-sequencing studies have identified mutations in the *p53* coding region in over 96% of high-grade serous ovarian carcinomas, the most malignant and common ovarian cancer subtype[10]. In addition to ovarian cancer, p53 mutations are also common in basal breast (88%), head and neck (57%), esophagus (43%), colon (43%), pancreatic (41%), and lung (37%) carcinomas[11–13]. Mutations in *p53* are believed to occur early in several cancers and have been shown to play key roles in tumorigenesis and development of drug resistance[1,14–16]. While some of these mutations contribute to cancer progression as a result of loss of wild-type (WT) p53 activity, many result in the gain of an oncogenic function[1,17]. These gain-of-function (GOF) oncogenic p53 mutant proteins (mutp53) accumulate to high levels in cells, form stable protein aggregates, activate alternative gene expression programs, and contribute to carcinogenesis as well as drug resistance[1,17]. Given their widespread presence in human cancer and key role in disease progression, targeting GOF mutp53 has emerged as an attractive therapeutic opportunity[1].

Increasing evidence indicates that the stabilization of mutp53 proteins is the key to their oncogenic activity[1,18]. Unlike WT-p53, which is rapidly degraded by the ubiquitin-proteasome system, the GOF mutp53 proteins, such as the p53-R175H, p53-R248Q, and p53-R273H are highly stable and have a tendency to form higher-order aggregates[1,18]. Depletion of GOF mutp53 in cells, harboring these mutations, induces cell death underscoring the merit of developing strategies that selectively target mutp53 in cancer cells[1,19,20]. However, the lack of precise understanding of the various factors that regulate their stability and turnover has impeded specific and selective targeting of mutp53 proteins in cancer cells.

In this report, we identify a previously unknown pathway that selectively regulates the p53-R175H GOF mutant protein. We show that a small-molecule compound called MCB-613, previously characterized as a steroid receptor coactivator (SRC) "super" stimulator, causes rapid and selective depletion of p53-R175H protein via an ubiquitin dependent lysosome-mediated pathway[21]. Using small molecule deubiquitinase (DUB) inhibitors and siRNA-mediated knockdown, we identify USP15 as a DUB that regulates p53-R175H levels in ovarian cancer cells. Taken together, our work demonstrates that distinct regulatory pathways and mechanisms dictate the stability, turnover of p53-WTm, and the different clinically important GOF mutp53, thereby opening new opportunities to selectively target them.

## Results

### MCB-613 causes rapid and selective depletion of p53-R175H.

We identified that a small-molecule compound called MCB-613 caused a rapid and sustained decrease in the level of the usually stable p53-R175H GOF mutant in the ovarian cancer cell line TYK-Nu (Fig. 1a, b and Supplementary Fig. 1A). Interestingly, in contrast to the effect on p53-R175H, a slight increase in the level of p53-WT protein was observed upon MCB-613 treatment in ALST cells (Fig. 1c). Furthermore, MCB-613 treatment had minimal effects on the other frequently observed GOF mutp53 (R248Q, R273H, and Y220C) in multiple cell lines (Fig. 1d,e and Supplementary Fig. 1B). To determine whether the effect of MCB-613 on p53-R175H mutant is specific to the ovarian cancer cell line TYK-Nu or mediated through a conserved mechanism,

we tested the effect of MCB-613 on p53-R175H in TOV-112D (ovarian cancer) and SK-BR-3 (breast cancer) cells. Similar to the results using TYK-Nu cells, MCB-613 treatment resulted in dramatic decrease in p53-R175H levels in both TOV-112D and SK-BR-3 cells (Fig. 1f,g). Consistent results were also observed using ectopically expressed p53-R175H, p53-R273H, and p53-WT in the p53-null SKOV3 cells upon MCB-613 treatment, further suggesting that the effect of the small molecule MCB-613 on the conformational p53-R175H mutant is mediated through a conserved mechanism (Fig. 1h).

Because the p53-R175H mutant is known to form aggregates in cells, we sought to confirm if MCB-613 caused reduction of p53-aggregates. Both immunofluorescence and native gel-western blot analyses revealed that MCB-613 treatment indeed resulted in a dramatic decrease in total p53-R175H protein levels, including aggregates (Fig. 1i and Supplementary Fig. 1C). To determine the effect of MCB-613 on other mutp53 proteins frequently observed in human cancers, and to ascertain whether the effect was specific to p53-R175H, we expressed frequently observed p53 mutants in SKOV3 cells, and evaluated the effect of MCB-613 on their levels. The effect of MCB-613 was found to be most profound on p53-R175H mutant protein. In addition to p53-R175H, the levels of C176Y, H179R, and H193R mutants also decreased slightly upon MCB-613 treatment (Fig. 1j). It is possible that mutations in the region near the R175 codon could potentially induce structural changes that are similar to those induced by R175H, and is important in mediating the response to MCB-613. Further, no noticeable effect was observed in the levels of the Y220C, S241F, R248Q, and C277H mutants upon MCB-613 treatment (Fig. 1j).

### Effect of MCB-613 on p53-R175H is independent of steroid receptor co-activators.

MCB-613 is a known small-molecule stimulator of the steroid receptor co-activators (SRCs), and also causes rapid turnover of SRC-1, SRC-2, and SRC-3 proteins in cells (Supplementary Fig. 2A)[21]. To ascertain if the effect of MCB-613 on p53-R175H is mediated through an SRC dependent transcriptional mechanism, we measured the changes in p53 transcript levels in ALST and TYK-Nu cells, upon MCB-613 treatment. The absence of correlation between the p53 mRNA and protein levels, upon MCB-613 treatment in both ALST and TYK-Nu cells, confirmed that the effect of MCB-613 on p53-WT and p53-R175H levels was independent of direct SRC dependent transcriptional regulation (Fig. 2a). MCB-613 also had no significant effect on the p53 mRNA levels in OVCAR3 (R248Q) and OVCA420 (R273H) cells (Supplementary Fig. 2B). To ascertain whether the effect on p53-R175H is dependent on SRCs, we used siRNA to knockdown SRC-1, SRC-2, or SRC-3 in TYK-Nu and ALST cells. The knockdown of SRC-1, SRC-2, or SRC-3 had no effect on either p53-WT or p53-R175H mutant protein levels (Fig. 2b–e). The SRC-1 levels, in ALST cells, were very low and undetectable by western blot analysis (Fig. 2b). Further, the small-molecule SRC-3 inhibitor, SI2, did not alter p53-R175H levels (Fig. 2f). Collectively, these results confirmed that the effect of MCB-613 on p53-WT and the p53-R175H mutant is independent of the transcriptional co-activators SRC-1, SRC-2, and SRC-3 in these cells.

### MCB-613 causes lysosome-mediated turnover of p53-R175H.

To determine whether p53-WT and the R175H mutant are regulated by MCB-613 at the translational level or through a post-translational mechanism, we treated ALST and TYK-Nu cells concurrently with MCB-613 and the protein synthesis inhibitor cycloheximide. Cycloheximide-chase analysis revealed that, MCB-613 causes a dramatic decrease in the half-life of the usually stable p53-R175H mutant protein (Fig. 2g). In contrast to the

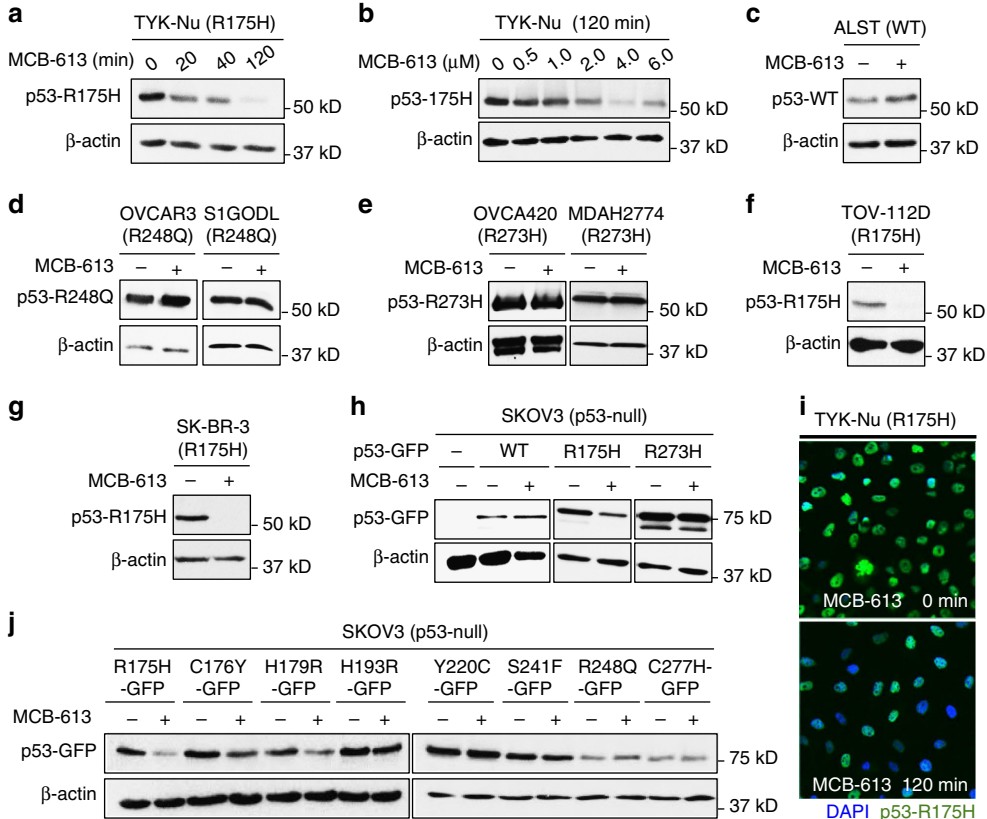

**Fig. 1** MCB-613 causes a rapid and selective decrease in the level of p53-R175H. **a** MCB-613 (6 μM) caused a rapid decrease in p53-R175H protein levels in TYK-Nu cells. **b** Effect of different concentrations of MCB-613 (2 h) on p53-R175H levels in TYK-Nu cells. MCB-613 treatment resulted in the decrease in p53-R175H levels in a concentration dependent manner. **c** p53-WT protein levels in ALST cells slightly increased upon MCB-613 (6 μM, 2 h) treatment. **d**, **e** MCB-613 (2 h) had minimal effect on other GOF mutp53, such as **d** p53-R248Q levels in OVCAR3 and S1GODL cells, and **e** p53-R273H levels in OVCA420 and MDAH2774 cells. **f**, **g** p53-R175H levels decreased in multiple cell lines upon MCB-613 (2 h) treatment; **f** TOV-112D (ovarian cancer); **g** SK-BR-3 (breast cancer). **g** Effect of MCB-613 (2 h) on GFP-tagged p53-WT, and p53-R175H mutant protein in the p53$^{-/-}$ SKOV3 cells. **h** Effect of MCB-613 (2 h) on GFP-tagged p53-WT, p53-R175H, and p53-R273H mutants in SKOV3 cells. **i** Immunofluorescence shows decrease in p53-R175H levels (green) in TYK-Nu cells upon MCB-613 treatment (2 h). **j** Effect of MCB-613 (2 h) on GFP-tagged p53 GOF mutants in SKOV3 cells

effect on R175H mutant, the half-life of p53-WT protein increased in the presence of MCB-613, resulting in its accumulation, indicating the effect of MCB-613 on p53-WT and R175H mutant to be post-translational (Fig. 2h). In comparison, MCB-613 had no effect on the turnover of p53-R248Q and p53-R273H proteins at these early-time points (Supplementary Fig. 2C, D). Because MCB-613 causes increased p53-R175H turnover, we sought to determine whether the effect is mediated via the proteasome or the lysosomal pathway. The proteasome inhibitor MG132 was unable to rescue the MCB-613 induced turnover of the p53-R175H mutant protein, suggesting that the protein is not targeted to the proteasome for degradation (Fig. 2i). In contrast, treatment of TYK-Nu cells with lysosomal inhibitors reversed the effect of MCB-613, thereby demonstrating that the drug causes selective turnover of the p53-R175H GOF mutant by targeting it to the lysosome (Fig. 2j). The autophagy inhibitor, LY294002, also reversed the effect of MCB-613 on p53-R175H, suggesting a potential role for autophagosomes in MCB-613-induced p53-R175H turnover (Supplementary Fig. 2E). Interestingly, MCB-613 was previously reported to induce the formation of autophagosomes in breast cancer cells[21].

Interestingly, MCB-613 causes rapid export of the p53-R175H mutant protein from the nucleus to the cytoplasm (Fig. 3a and Supplementary Fig. 3A). No change in p53 localization was observed in cells expressing p53-WT or the R248Q and R273H mutants (Fig. 3b–d and Supplementary Fig. 3B). Further, consistent with being targeted to the lysosome for degradation,

p53-R175H localized to the lysosome upon MCB-613 treatment (Fig. 3e). We also observed co-localization of p53-R175H with calnexin, an endoplasmic reticulum (ER) marker (Fig. 3f). Although the significance of ER localization is not entirely clear, we hypothesize that localization to ER might serve as an intermediate step in targeting the R175H mutant protein to the lysosome for degradation, in response to MCB-613. No significant localization of the R175H mutant was observed with Mitotracker, a mitochondria-specific dye (Supplementary Fig. 3C).

**MCB-613 treatment causes rapid ubiquitination of p53-R175H.** A short MCB-613 treatment (<20 min) results in the accumulation of higher migrating p53 isoforms in TYK-Nu cells (R175H), but not in ALST (WT) or OVCA420 (R273H) cells as shown by western blot analyses (Fig. 4a). To ascertain if these higher molecular-weight bands represent ubiquitinated forms of the R175H mutant, we immunoprecipitated mutp53 protein from the control and MCB-613 treated TYK-Nu cells, and performed an anti-ubiquitin western blot. A marked increase in ubiquitinated p53-R175H protein was observed upon MCB-613 treatment (Fig. 4b). To confirm this result, we ectopically expressed His-tagged ubiquitin (His-Ub) in TYK-Nu cells, and performed Ni-NTA pull down of cellular proteins containing His-Ub from the control and MCB-613 treated cells, followed by an anti-p53 immunoblot. Consistent with our previous result, MCB-613

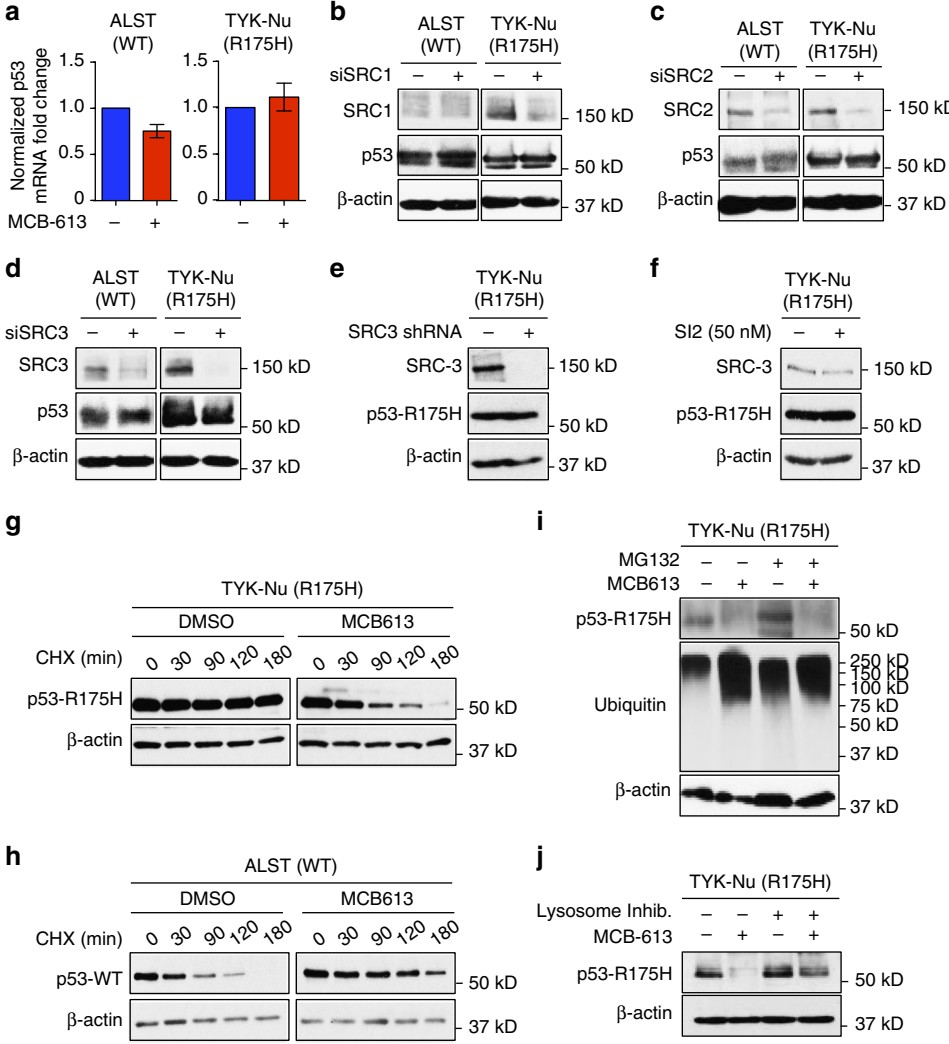

**Fig. 2** MCB-613 causes lysosome-mediated degradation on p53-R175H. **a** Effect of MCB-613 (2 h) on p53 mRNA levels in ALST and TYK-Nu cells. Values are normalized mean ± s.e.m. ($n = 3$). **b–d** siRNA-mediated knockdown (48 h) of SRC-1 (**b**), SRC-2 (**c**), and SRC-3 (**d**) had no effect on p53-WT (ALST cells) and p53-R175H (TYK-Nu cells) levels. **e** shRNA-mediated knockdown of SRC3 (72 h) in TYK-Nu cells does not affect p53-R175H levels. **f** SRC-3 inhibitor SI2 (50 nM, 2 h) has no effect on p53-R175H levels in TYK-Nu cells. **g, h** MCB-613 caused a **g** decrease in the half-life of p53-R175H and **h** increase in the half-life of p53-WT. **i** Proteasome inhibitor MG132 does not inhibit MCB-613 induced turnover of p53-R175H. **j** Lysosome inhibitors (pepstatin A, Leupeptin, and E-64D) rescued MCB-613 induced turnover of p53-R175H

treatment results in increased pull down of ubiquitinated p53-R175H protein (Supplementary Fig. 4A). Increased ubiquitination of p53-R175H mutant protein, in response to MCB-613 treatment, is also observed in SKOV3 cells expressing ectopic p53-R175H and His-Ub, suggesting the mechanism to be conserved across different cell types (Fig. 4c). The levels of both K-48 and K-63-linked ubiquitin chains are enhanced on p53-R175H, in response to MCB-613 treatment (Supplementary Fig. 4B and C). However, the increase in K-63-linked ubiquitin chain appeared to be more dominant, consistent with lysosomal export of the ubiquitinated protein (Supplementary Fig. 4C). To identify the regions within p53-R175H protein, important in mediating sensitivity to MCB-613, we expressed N and C-terminal truncated versions of p53-R175H in SKOV3 cells. Deletion of the N-terminal 42 amino acids (43–393) or C-terminal 30 amino acids (1–363) made p53-R175H resistant to MCB-613-induced degradation (Supplementary Fig. 4D). WT p53 has 20 lysine residues that can be potentially ubiquitinated (Supplementary Fig. 4E). To determine if any of the lysines within the N-terminal or C-terminal region, identified to mediate sensitivity to MCB-613,

is critical for the response, we used site-directed mutagenesis to generate lysine-to-arginine mutants and expressed them in SKOV3 cells. None of the single or double lysine substitution mutants we tested, resulted in resistance to MCB-613 induced turnover of the R175H mutant (Supplementary Fig. 4F). The inability of the tested lysine substitution mutants to resist MCB-613 induced degradation suggests: (i) possible redundancy in ubiquitination sites on p53-R175H or (ii) that the C-terminal and N-terminal regions do not harbor the critical ubiquitination sites, but are important for mediating protein–protein interactions necessary for MCB-613 induced ubiquitination and turnover of the R175H mutant.

**MDM2 mediates the effect of MCB-613 on p53-WT.** Previous studies have shown that MCB-613 causes oxidative stress in cells[21]. To determine whether MCB-613-induced oxidative stress mediates the effect on p53-WT and R175H mutant, we treated TYK-Nu and ALST cells with increasing concentrations of $H_2O_2$ for 1 h. While p53-WT levels decreased slightly at higher $H_2O_2$

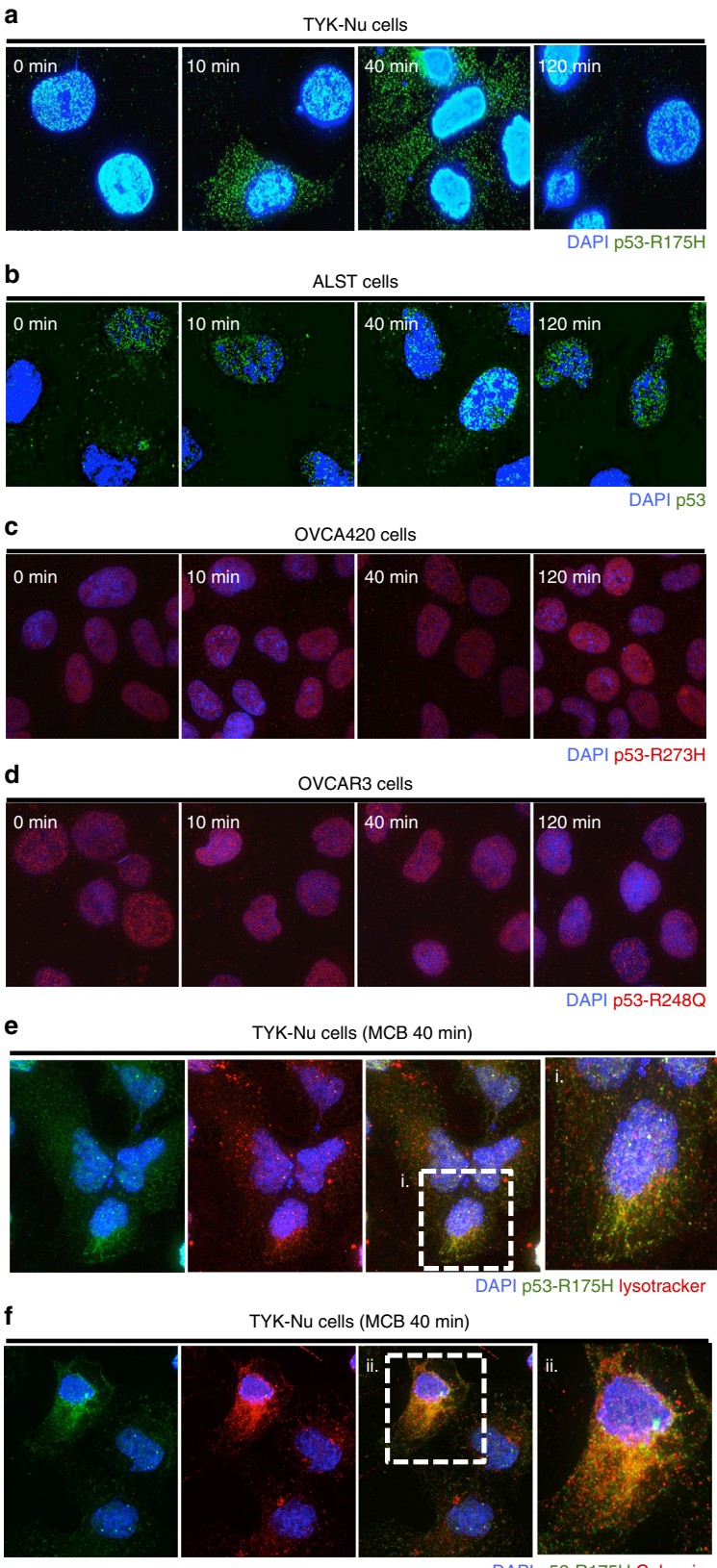

**Fig. 3** MCB-613 causes nuclear export of p53-R175H protein. **a**–**d** MCB-613 caused **a** rapid export of p53-R175H (TYK-Nu cells) protein but not **b** p53-WT (ALST cells) or other GOF mutp53 **c** p53-R273H (OVCA420 cells), and **d** p53-R248Q (OVCAR3 cells). **e**, **f** Upon MCB-613 treatment, p53-R175H (green) co-localizes with **e** lysosomes (red; lysotracker) and **f** endoplasmic reticulum (red; α-calnexin) in TYK-Nu cells

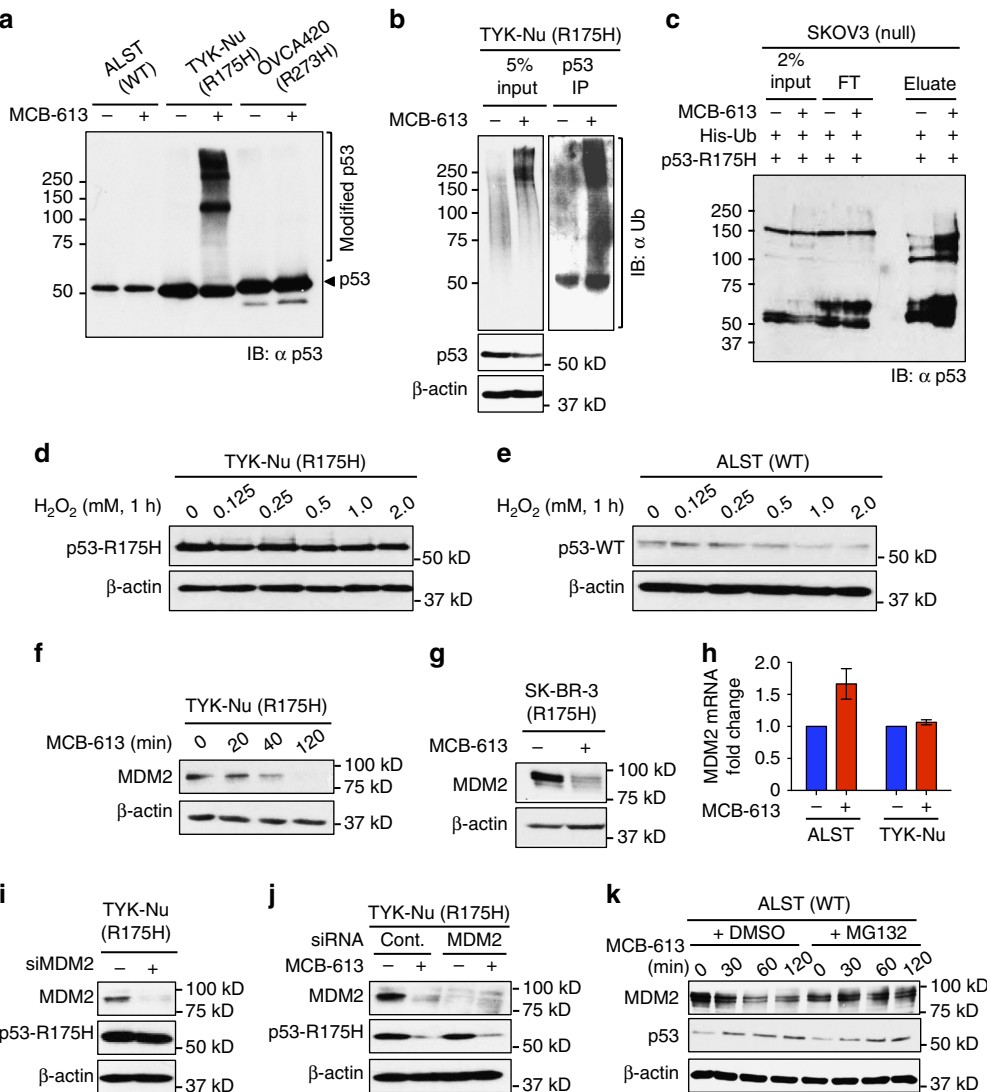

**Fig. 4** MCB-613 causes increased ubiquitination of p53-R175H protein. **a** Higher migrating p53-R175H bands accumulate in TYK-Nu cells upon short-term (<20 min) MCB-613 treatment; but not in ALST (p53-WT) or OVCA420 (p53-R273H) cells. **b** MCB-613 caused increased ubiquitination of p53-R175H in TYK-Nu cells. **c** Increased ubiquitination of ectopic p53-R175H-HA upon MCB-613 treatment in SKOV3 cells. **d** Effect of increasing concentrations of $H_2O_2$ (1 h) on p53-R175H levels in TYK-Nu cells. **e** Effect of increasing concentrations of $H_2O_2$ (1 h) on p53-WT levels in ALST cells. **f, g** MCB-613 causes decrease in MDM2 levels in **f** TYK-Nu and **g** SKBR-3 cells. **h** Effect of MCB-613 on MDM2 mRNA levels in ALST and TYK-Nu cells. Values are normalized mean ± s.e.m. ($n = 3$). **i** MDM2 knockdown (48 h) has no effect on p53-R175H levels in TYK-Nu cells. **j** Effect of MCB-613 on p53-R175H turnover is independent of MDM2. **k** MG132 reversed MCB-613 induced MDM2 turnover in ALST cells

concentrations, $H_2O_2$ has no effect on p53-R175H levels (Fig. 4d, e). These results indicate that the effect of MCB-613 on p53-WT and R175H mutant is not mediated by cellular oxidative stress. WT-p53 levels in cells are tightly regulated by the ubiquitin-proteasome system. The E3 ligase MDM2 is known to ubiquitinate p53-WT and target it for degradation. Although the role of MDM2 in p53-WT degradation is well-established, its role in regulating the turnover of the p53-R175H mutant is not clear[22]. Recent reports have identified several small molecules that can bind mutp53 and induce conformational changes that make the p53 mutants assume a WT-like conformation (a process termed "reactivation"), and thereby become a target of MDM2-mediated turnover[1]. To determine if MCB-613 operates through such a mechanism, we tested the ability of nutlin-3A to reverse the effect of MCB-613 on p53-R175H in TYK-Nu cells. Nutlin-3A has been shown to stabilize p53-WT by disrupting p53-MDM2 interaction[23]. Nutlin-3A did not reverse the effect of MCB-613 on p53-R175H, thereby confirming that MCB-613 does not cause

reactivation of p53-R175H (Supplementary Fig. 4G). Consistently, we found that MCB-613 does not bind to recombinant p53-R175H protein (Supplementary Fig. 4H). Interestingly, MCB-613 causes a rapid decrease in MDM2 levels in multiple cell lines (Fig. 4f, g, k). The effect of MCB-613 on MDM2 was not transcriptional (Fig. 4h). MDM2 mRNA slightly increased in ALST cells upon MCB-613 treatment, possibly due to the transcriptional feedback from elevated p53-WT protein in these cells (Fig. 4h). No change in MDM2 mRNA was observed upon MCB-613 treatment in TYK-Nu cells (Fig. 4h). The decrease in MDM2 protein levels would explain the stabilization of p53-WT in ALST cells, but we wanted to determine if MDM2 mediated the decrease in p53-R175H stability. siRNA knockdown of MDM2 had no effect on the p53-R175H levels (Fig. 4i). Further, MCB-613 caused a dramatic reduction in p53-R175H levels even in the absence of MDM2 in both TYK-Nu cells and $p53^{-/-}$:$MDM2^{-/-}$ MEFs, expressing ectopic p53-R175H (Fig. 4j and Supplementary Fig. 4I). Collectively, these results demonstrate that MDM2

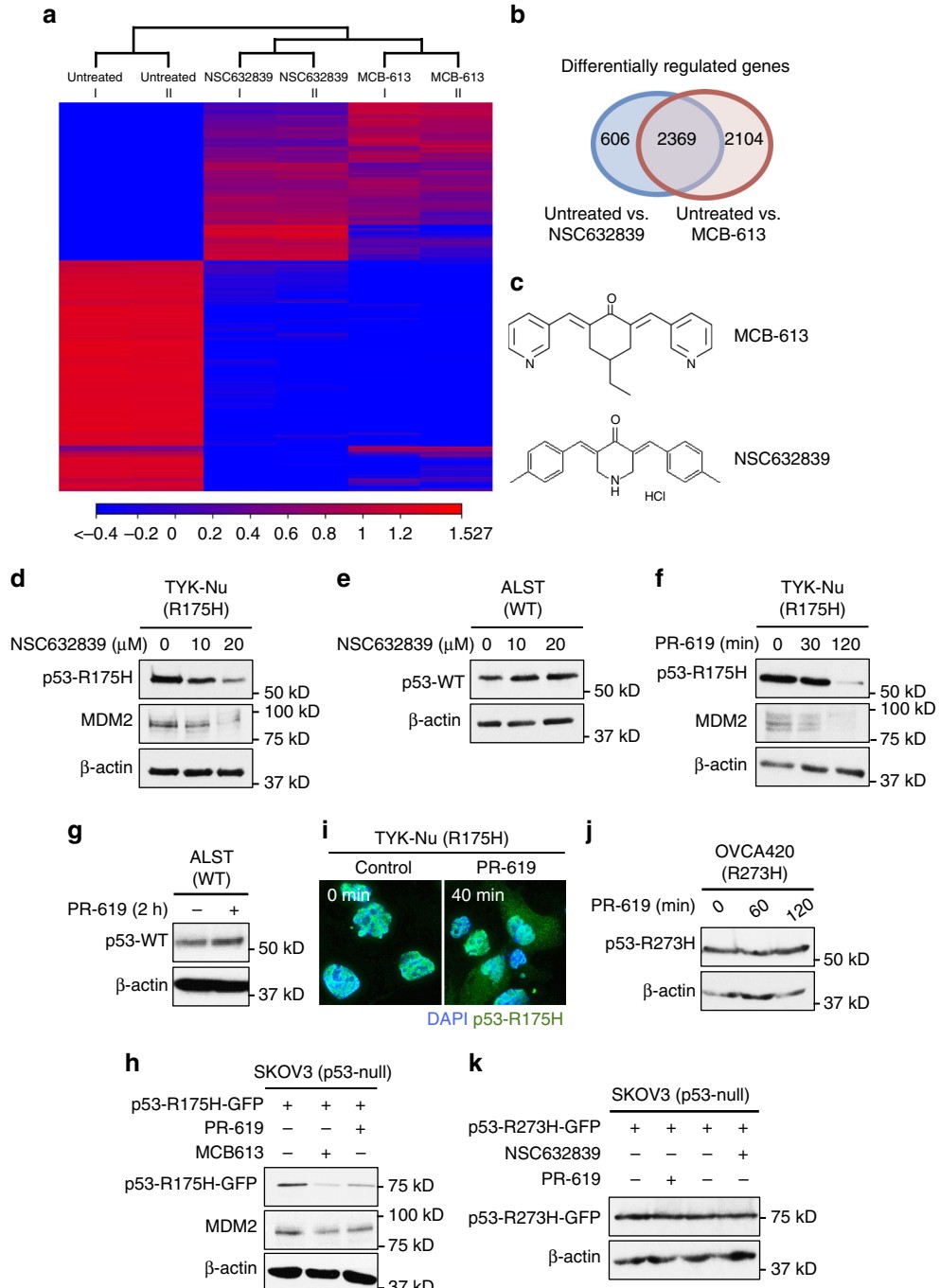

**Fig. 5** DUB inhibitors mimic the effect of MCB-613 on p53-WT and p53-R175H. **a** Comparison of RNAseq data from MCF-7 cells treated with MCB-613 (6 h) and the DUB inhibitor NSC632839. **b** Venn diagram summarizing the result in Fig. 5a shows that the gene-expression changes induced by MCB-613 (6 h) and the DUB inhibitor NSC632839 in MCF-7 cells are highly similar. **c** Chemical structure of MCB-613 and NSC632839. **d**, **e** NSC632839 (20 μM, 2 h) causes **d** depletion of p53-R175H and MDM2 and **e** slight increase in p53-WT levels. **f**, **g** PR-619 (10 μM, 2 h) causes **f** depletion of p53-R175H and MDM2 and **g** slight increase in p53-WT levels. **h** Effect of MCB-613 (6 μM, 2 h) and the pan-DUB inhibitor PR-619 (10 μM, 2 h) on ectopically expressed GFP tagged p53-R175H protein and endogenous MDM2 in SKOV3 cells (p53-null). **i** PR-619 causes cytoplasmic localization of p53-R175H (green). **j** Effect of PR-619 (10 μM) on p53-R273H mutant protein in OVCA420 cells. **k** Effect of DUB inhibitors PR-619 (10 μM, 2 h) and NSC632839 (20 μM, 2 h) on ectopically expressed GFP-tagged p53-R273H in the p53-null SKOV3 cells

mediates the effect of MCB-613 on p53-WT, but the effect on p53-R175H is independent of MDM2. Moreover, MCB-613 caused increased ubiquitination of MDM2 (Supplementary Fig. 4J). To determine whether MDM2 is targeted to the proteasome for degradation upon MCB-613 treatment, we treated ALST cells simultaneously with both MCB-613 and proteasome inhibitor MG132. Concurrent MG132 treatment rescued

MCB-613 induced MDM2 degradation, suggesting that MCB-613 causes proteasome-mediated turnover of MDM2 in ALST cells (Fig. 4k).

**DUB inhibitors mimic the effect of MCB-613 on p53-R175H.** Since MCB-613 treatment resulted in increased ubiquitination of

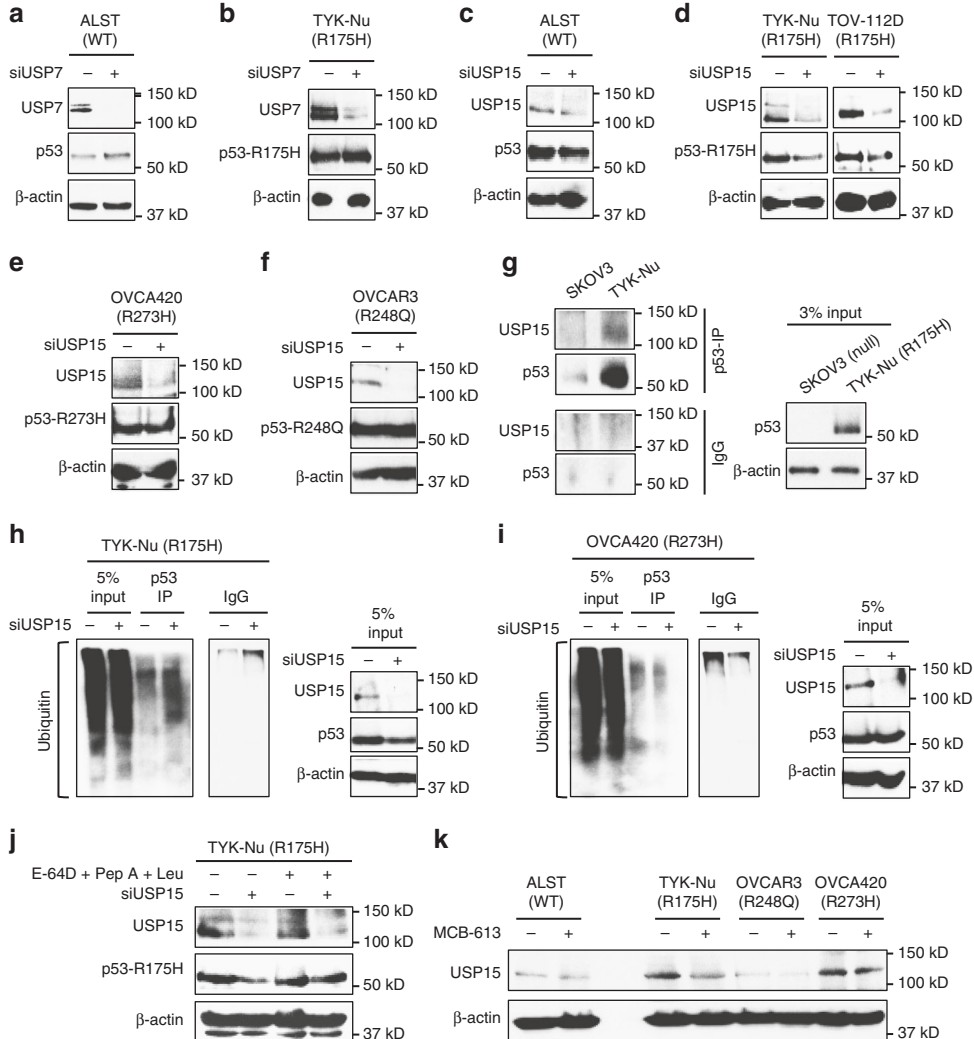

**Fig. 6** USP15 regulates p53-R175H levels in ovarian cancer cells. **a**, **b** Effect of USP7 knockdown on **a** p53-WT and **b** p53-R175H levels. **c**, **d** USP15 knockdown causes **c** no effect on p53-WT levels (ALST cells) but **d** decrease in p53-R175H levels (TYK-Nu and TOV-112D cells). **e**, **f** siRNA-mediated knockdown of USP15 has no effect on the levels of **e** p53-R273H in OVCA420 cells and **f** p53-R248Q in OVCAR3 cells. **g** USP15 co-immunoprecipitates with p53-R175H. **h** Increased ubiquitination of p53-R175H upon USP15 knockdown. p53-R175H was immunoprecipitated from TYK-Nu cells treated with non-targeting siRNA and USP15 siRNA, using anti-p53 antibody (FL-393). Immunoblot for ubiquitin (anti-Ub (P4D1)) was performed on the immunoprecipitated proteins. **i** USP15 knockdown by siRNA does not result in increased ubiquitination of p53-R273H in OVCA420 cells. **j** Lysosomal inhibitors reversed the effect of USP15 siRNA on p53-R175H levels. **k** MCB-613 treatment (6 μM, 2 h) results in decrease in USP15 protein levels

p53-R175H, we hypothesized that MCB-613 might be acting through pathways that either augment the activity of specific ubiquitin ligases, or inhibit the activity of DUBs. Connectivity map (CMap) analysis of the RNAseq data, from MCF-7 cells treated with MCB-613 with the publically available RNAseq data post-treatment with other chemical compounds, revealed remarkable functional similarity between MCB-613 and a structurally similar DUB inhibitor, NSC632839 (Fig. 5a–c). We tested the ability of NSC632839 to mimic the effect of MCB-613 on p53-R175H and p53-WT. Similar to MCB-613, NSC632839 treatment results in the depletion of p53-R175H and a slight increase in p53-WT (Fig. 5d,e). NSC632839 treatment also causes a decrease in MDM2 levels (Fig. 5d). Further, the reversible cell permeable pan-DUB inhibitor PR-619 also causes a decrease in p53-R175H and MDM2 levels in TYK-Nu cells (Fig. 5f). Consistent with previous results using MCB-613 and NSC632839, an increase in p53-WT was observed upon PR-619 treatment (Fig. 5g). Similar to MCB-613, the effect of PR-619 was also conserved in SKOV3 cells expressing ectopic p53-R175 H (Fig. 5h). Again, similar to MCB-613, PR-619 causes export of nuclear p53-R175H protein

into the cytoplasm (Fig. 5i). In contrast to its effect on p53-R175H and p53-WT, PR-619 and NSC632839 did not alter the levels of p53-R273H mutant in either OVCA420 cells or SKOV3 cells expressing GFP-tagged p53-R273H (Fig. 5j,k), suggesting the effect is selective to p53-R175H. Unlike PR-619 and NSC632839, the USP14 inhibitor IU-1 did not alter the levels of either p53-WT or the R175H mutant, suggesting specificity in DUBs involved in mediating the stability and turnover of p53-WT and the R175H mutant (Supplementary Fig. 5A).

**USP15 regulates the stability of p53-R175H GOF mutant.** Because the DUB inhibitor NSC632839 successfully mimicked the response elicited by MCB-613 treatment on p53-WT and the R175H mutant, we decided to test the role of USP7, a known NSC632839 targeted DUB[24], on the stability and turnover of p53-WT and p53-R175H mutant. siRNA-mediated knockdown of USP7, in ALST cells, resulted in elevated p53-WT levels similar to those observed with MCB-613 and DUB inhibitors (Fig. 6a). This result is consistent with the previously known role of USP7 in regulating p53-WT stability[25]. However, USP7 knockdown in

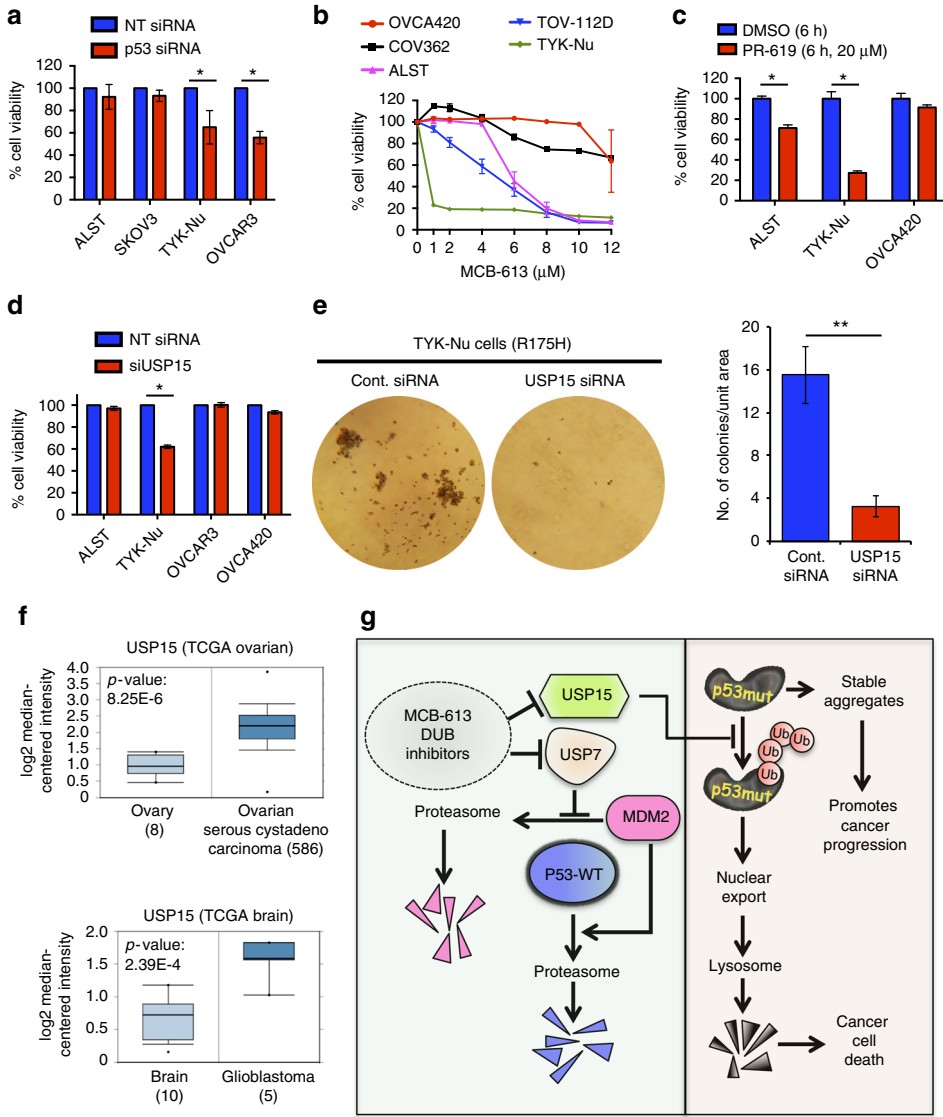

**Fig. 7** USP15 depletion causes cancer cell death in ovarian cancer cells expressing p53-R175H. **a** p53 knockdown causes decreased viability in ovarian cancer cells expressing GOF mutp53, but not p53-WT and p53-null cells. Values are normalized mean ± s.e.m. ($n = 3$; *$p$-value < 0.05). **b** Ovarian cancer cells expressing p53-R175H (TYK-Nu and TOV-112D) are more sensitive to MCB-613 than ALST (p53-WT), OVCA420 (p53-R273H), and COV362 (p53-Y220C). Values are normalized mean ± s.e.m. ($n = 3$). **c** TYK-Nu cells are more sensitive to PR-619 compared with ALST and OVCA420 cells. Values are normalized mean ± s.e.m. ($n = 3$; *$p$-value < 0.05). **d** Effect of USP15 knockdown on cell viability in ovarian cancer cells carrying different p53 mutation status. Values are normalized mean ± s.e.m. ($n = 3$; *$p$-value = 0.0164). **e** USP15 knockdown significantly reduced the anchorage-independent growth of TYK-Nu cells (p53-R175H). Values are normalized mean ± s.d. ($n = 4$; **$p$-value = 0.006). **f** Data retrieved from Oncomine showing elevated *USP15* mRNA levels in multiple cancers from previously published datasets[29, 39–42]. **g** Schematic showing that different pathways regulate the stability of p53-R175H and p53-WT in ovarian cancer cells. MCB-613 and DUB inhibitors, such as NSC632839 and PR-619 caused selective depletion of the GOF p53 mutant p53-R175H, while causing slight increase in p53-WT levels. While inhibition of USP15 by these small molecules or siRNA resulted in increased ubiquitination and lysosome-mediated turnover of p53-R175H, it had no effect on p53-WT levels. In contrast, USP7 depletion caused MDM2-mediated stabilization of p53-WT protein

TYK-Nu cells did not result in any noticeable reduction in p53-R175H levels (Fig. 6b). This suggested that the effect of DUB inhibitors on p53-WT and p53-R175H is mediated through potentially different DUBs. USP15 was recently described as capable of reducing MDM2 levels in cells[26]. Since PR-619 is known to inhibit USP15[27] (Supplementary Fig. 6C), we tested the effect of USP15 knockdown on p53-WT and p53-R175H levels. In contrast to previously reported effects on p53-WT[26], siRNA-mediated knockdown of USP15 did not cause any change in p53-WT levels in ALST cells (Fig. 6c). Interestingly, knockdown of USP15 in both TYK-Nu and TOV112D cells caused p53-R175H levels to decrease (Fig. 6d and Supplementary Fig. 6A). USP15

knockdown did not alter the levels of p53-R273H in OVCA420 cells or p53-R248Q in OVCAR3 cells (Fig. 6e, f). These results show that WT and the different mutp53 proteins are regulated differentially by DUBs in cancer cells, and establish USP15 as an upstream regulator of p53-R175H in ovarian cancer cells. Next, we performed p53-immunoprecipitation, from the p53-R175H mutant expressing TYK-Nu cells, to determine whether p53-R175H interacted with USP15. The p53-null SKOV3 cells served as a control for non-specific antibody interactions. USP15 was found to co-immunoprecipitation (co-IP) with p53-R175H in TYK-Nu cells (Fig. 6g). No interaction was observed by co-IP between USP15 and p53-WT or other GOF mutants (p53-R248Q

and p53-R273H) (Supplementary Fig. 6B). Since MCB-613 treatment induced increased ubiquitination of p53-R175H, we tested the ability of USP15 to mimic this response. p53-R175H was immunoprecipitated from TYK-Nu cells treated with either non-targeting siRNA or USP15 siRNA. Anti-ubiquitin immunoblotting of p53-IP revealed increased ubiquitination of p53-R175H upon USP15 knockdown (Fig. 6h). In contrast, USP15 knockdown did not mediate increased ubiquitination of p53-R273H in OVCA420 cells (Fig. 6i). Further, the depletion of p53-R175H upon USP15 knockdown was rescued by co-treatment with lysosomal inhibitors (pepstatin-A, leupeptin, and E-64D), suggesting that similar to MCB-613, USP15 causes selective depletion of p53-R175H through an ubiquitin-mediated lysosomal pathway (Fig. 6j).

To determine how MCB-613 might be interfering with USP15 function in cells, we first tested the possibility that MCB-613 functions as a direct inhibitor of USP15. However, unlike PR-619, MCB-613 did not cause a decrease in USP15 activity using the artificial substrate Ub-AMC in the in vitro DUB activity assay (Supplementary Fig. 6C and D). Further, no change in USP15 mRNA levels was observed upon MCB-613 treatment (Supplementary Fig. 6E). Interestingly, MCB-613 treatment resulted in decreased USP15 protein levels in TYK-Nu (R175H) and OVCA420 (R273H) cells, suggesting that MCB-613 causes depletion of USP15 protein through a post-translational mechanism (Fig. 6k and Supplementary Fig. 6F). These results are consistent with our previous observations that the effect of MCB-613 on p53-R175H is post-translational, and that depletion of USP15 causes decreases in p53-R175H levels in TYK-Nu cells, but not p53-R273H in OVCA420 cells.

**USP15 knockdown reduced viability of p53-R175H cancer cells.** GOF mutp53 proteins play important roles in the growth and survival of cancer cells, and selective depletion of mutp53 has been suggested as a potential strategy to induce targeted killing of cancer cells[1]. As expected, knockdown of mutp53 in ovarian cancer cells resulted in decreased cell viability (Fig. 7a and Supplementary Fig. 7A)[28]. Since MCB-613 causes selective depletion of p53-R175H mutant protein, we determined the sensitivity of ovarian cancer cell lines, expressing the different mutp53 to MCB-613. Both TYK-Nu and TOV-112D cells express the p53-R175H mutant, and were found to be more sensitive to MCB-613 treatment compared with ALST (p53-WT), OVCA420 (p53-R273H), and COV362 (p53-Y220C) cells (Fig. 7b). Increased sensitivity of TYK-Nu cells to cell death was also observed with the DUB inhibitor PR-619 (Fig. 7c).

Further, USP15 knockdown resulted in decreased cell viability in TYK-Nu cells, but not in ALST, OVCAR3, or OVCA420 cells (Fig. 7d). In contrast, USP7 knockdown did not have any effect of TYK-Nu cell viability (Supplementary Fig. 7B). Colony forming assays revealed a significant decrease in the ability of TYK-Nu cells to form colonies upon USP15 knockdown, and colonies that did form were small (Fig. 7e). The USP15 gene is amplified in several cancers including ovarian, glioblastoma, and breast cancer[29]. Publically available data accessible through Oncomine show that USP15 expression is significantly elevated in ovarian serous cystadenoacrcinoma, lobular breast carcinomas, prostate cancer, cervical squamous cell carcinomas, and glioblastomas, suggesting potential clinical relevance for this protein are these cancers, especially in relation to the p53 status in these tumors (Fig. 7f and Supplementary Fig. 7C)[29]. Thus, taken together, this study implicates USP15 as a previously unknown clinically important regulator of p53-R175H mutant protein in ovarian cancer cells, and further illustrates that distinct pathways regulate the levels and stability of the different mutp53 proteins and WT-p53 in cells (Fig. 7g).

## Discussion

Experiments using cancer cell lines and mouse models show that targeted depletion of GOF mutp53 in cancer cells causes decreased cell viability and proliferation leading to tumor regression[1]. While the activation of WT p53 and the knockdown of mutp53 can kill cancer cells expressing these proteins, the knockdown of WT-p53 in normal cells can predispose cells to cancer. Therefore, strategies that can distinguish between mutant and WT-p53 and achieve targeted depletion of mutp53 have been recognized to have high therapeutic potential. In this report, we demonstrate that distinct pathways can regulate mutant versus WT protein levels in ovarian cancer cells.

The regulation of p53-WT has been studied extensively[30]; most commonly it is ubiquitinated by MDM2, a well-established E3-ubiquitin ligase that targets WT p53 for degradation by proteasome. In addition, the stability of WT p53 is regulated by the DUB, USP7[25]. In contrast, very little is known about factors and pathways that regulate the stability and turnover of the pathologically important mutp53 proteins. Recently, statins were shown to cause the degradation of p53-R175H mutant protein (with minimum effects on p53-WT), an effect mediated through the mevalonate pathway, ubiquitin ligase CHIP and the HSP40 member DNAJA1[31]. Other reports describe small molecules and peptides that directly bind to and revert the conformation of mutp53 to a more WT-like form, a process termed "reactivation", thereby enabling their detection and turnover by mechanisms that regulate the rapid degradation of WT-p53[1].

In this report, we show that a small molecule called MCB-613 elicits opposite effects on p53-WT and p53-R175H mutant protein levels in ovarian cancer cells, and that these effects of MCB-613 on p53 turnover are independent of its role as a pan-SRC "super" stimulator[21]. Specifically, MCB-613 treatment resulted in a rapid and selective turnover of the p53-R175H conformational mutant, whereas it increased the half-life of p53-WT (Fig. 7g). While H193R, Y220C, S241F, R248Q, R273H, and C277H mutants were unaffected upon MCB-613 treatment, the levels of C176Y and H179R mutants decreased upon MCB-613 treatment, albeit to a lesser extent than observed for R175H, indicating that mutations around the R175 locus in the protein confer specific conformational sensitivities.

We also provide evidence to show that MCB-613 induces the specific effects of WT versus mutant p53 proteins, by potentially interfering with the levels and activity of specific DUBs, a previously uncharacterized mode of action for this promising investigational small molecule drug. Because MCB-613 causes the levels of the E3-ubiquitin ligase MDM2 to decrease, the MCB-613-mediated increase in p53-WT is likely to be mediated through MDM2[32] as well as by the DUB, USP7, as confirmed herein[25]. By contrast, the MCB-613-mediated turnover of p53-R175H is MDM2 and USP7 independent. The lack of dependence on MDM2 together with the inability of nutlin-3A to rescue MCB-613-induced turnover of p53-R175H suggests that MCB-613 is not causing turnover of the R175H mutant, by inducing changes in its conformation to a more WT-like form, as shown by other studies[33]. Rather, we provide evidence that MCB-613 is regulating p53 R175H degradation, by distinctly different mechanisms that involve specific ubiquitination, nuclear export, and targeted degradation in lysosomes, not proteasomes. Although consistent and reproducible, the effect of lysosome inhibitors on rescuing MCB-613 induced p53-R175H turnover was not 100%. It could be potentially due to: (1) the effectiveness of these inhibitors and (2) the kinetics of lysosome inhibitors compared with the effect of MCB-613 on p53-R175H (which is very rapid), and (3) in the absence of any evidence against it, we also cannot rule of possibility of other potential mechanisms. We identify USP15 as a specific DUB that regulates the levels of p53-

R175H mutant, but not p53WT protein. Considering the existing redundancy in structure and function between various DUBs, evaluating the roles of other DUBs in regulating mutp53 proteins should provide additional insights into these complex regulatory pathways controlling protein stability and turnover. Taken together, this study identifies USP15 as a DUB targeting the oncogenic GOF p53-R175H mutant and provides evidence toward the existence of distinct pathways that regulate the stability and turnover of p53-WT and the different mutp53 proteins.

Although we identified USP15 as one of the DUBs regulating p53-R175H, the identity of the E3 ligase(s) targeting p53-R175H remains to be identified. Our efforts to identify specific lysine residues on p53-R175H, critical in mediating the response to MCB-613, were not successful likely because multiple lysine residues on p53-R175H are ubiquitinated in response to MCB-613, and because there is redundancy in ubiquitination sites on p53[34]. These questions are subjects of future investigations and should reveal valuable details on how p53-R175H is regulated in cells.

*USP15* is amplified in several cancers, including ovarian cancer[29]. Gene expression data available through Oncomine reveal elevated *USP15* expression in ovarian serous cystadenoacrcinoma, lobular breast carcinomas, prostate cancer, cervical squamous cell carcinomas, and glioblastomas (Fig. 7f and Supplementary Fig. 7C). Depletion of USP15 represses the oncogenic ability of patient-derived glioma-initiating cells[29]. We show that, USP15 knockdown resulted in a selective decrease in cell viability of ovarian cancer cells expressing the p53-R175H mutant protein. USP15 deficiency also promotes T-cell activation and enhances immune responses to tumors in specific mouse models[26]. Usp15 KO mice have higher levels of effector T-cells, that infiltrate tumors and promote resistance to transplanted tumor growth[26]. Thus targeting USP15 in tumors containing p53-R175H mutant appears to provide the dual advantage of targeting oncogenic mutp53 and enhancing cancer immunotherapy. Ultimately, strategies that enable selective depletion of oncogenic mutp53 protein will pave way for effective personalized treatment of cancer patients, depending on their p53 mutation status.

## Methods

**Cell lines and cell culture.** HEK293T, SKOV3, and SK-BR-3 cells were purchased from ATCC. TYK-Nu cells were a kind gift from Dr. Hilary Kenny, University of Chicago. ALST, OVCAR3, OVCA420, S1GODL, MDAH2774, COV362, and TOV-112D were provided by Dr. KK Wong (MD Anderson Cancer Center). HEK-293T cells were grown in DMEM media containing L-glutamine and supplemented with 10% FBS. All ovarian cancer cells were cultured in RPMI-1640 media containing L-glutamine supplemented with FBS (10%), MEM vitamins (1%), and non-essential amino acids (1%). SK-BR-3 cells were grown in McCoy's 5A modified media supplemented with FBS (10%). The p53 mutations in these cell lines were confirmed as described previously[28]. p53$^{-/-}$: MDM2$^{-/-}$ MEFs (a kind gift from Dr. Gigi Lozano (MD Anderson Cancer Center, USA)) were cultured in DMEM containing 10% FBS. Standard cell culture techniques were followed while culturing these cells. All cells were verified to be mycoplasma-free and maintained in a 37 °C humidified atmosphere with 5% CO$_2$.

**Plasmids and reagents.** Mammalian cell expression as p53-WT and mutants were achieved by cloning them into pEGFP-N1 vector as described previously[28]. p53-R175H truncation-mutant plasmids were a kind gift from Dr. Wenwei Hu (Rutgers Univ., USA). pEGFP-p53-R175H lysine-to-arginine mutant plasmids were developed by site-directed mutagenesis using respective primers (Supplementary Table 1). Reagents were obtained as indicated: cycloheximide (Sigma), MG132 (Sigma), E-64D (Sigma), pepstatin-A (Sigma), leupeptin (Sigma), lysotracker (Life Technologies), mitotracker (Life Technologies), MCB-613 (provided by Dr. David Lonard, BCM), PR-619 (Cayman chemicals), F6 or NSC632839 (Cayman chemicals), and IU-1 (Cayman chemicals). List of antibodies used in this study and dilutions are described in Supplementary Table 2.

**Plasmid transfection and gene knockdown.** For transfection of plasmid DNA and gene knockdown experiments, cells were grown to 70–80% confluency. Attractene (Qiagen) was used to transfect plasmid DNA as per the manufacturer's instructions. Dharmafect (Dharmacon) was used to transfect siRNA (ON-

TARGET plus Human siRNA-SMART pool, Dharmacon). SRC3 shRNA in GIPZ lentiviral vector was used to make lentivirus particles in HEK293T cells. USP15 shRNA 1 (clone ID:NM_006313.1-3391s1c1) and USP15 shRNA 2 (clone ID: NM_006313.1-1346s1c1) were ordered from Sigma. Ovarian cancer cells were infected with lentivirus particles for 72 h to effect shRNA knockdown.

**Details of drug treatments.** MCB-613 was solubilized in DMSO, and cells were treated at a concentration of 6 μM unless specified otherwise. PR-619 was treated at a concentration of 10 μM. NSC632839 was treated at a concentration of 20 μM. IU-1 was used at a concentration of 50 μM. To determine effect of proteasome and lysosomal inhibition on p53-R175H levels in MCB-613 treated cells, the cells were pre-treated with either the proteasome inhibitor (MG132 (10 μM)) or the lysosomal inhibitors (pepstatin A (10 μg/ml), leupeptin (200 μM), and E-64D (10 μg/ml)) for 20 min prior to MCB-613 treatment (1 h). Cycloheximide was used at a concentration of 25 μg/ml.

**Quantitative PCR analysis.** Total RNA from cells was extracted using the RNeasy Kit (Qiagen), according to the manufacturer's instructions. Purified total RNA was reverse-transcribed using the Superscript cDNA synthesis kit (Bio-Rad). Quantitative PCR was performed in triplicates as per the instructions in the QuantinivaTM SYBr Green PCR kit (Qiagen), using the Qiagen Rotor Gene Q qPCR machine. Fold difference in gene expression was determined after normalizing against RPL19 mRNA. Error bars are ± standard error of three independent experiments. Primers used for qRT-PCR are listed in Supplementary Table 3.

**Western blot analysis.** Western blot analysis was performed as described previously[35,36]. Briefly, cells were lysed in lysis buffer (150 mM NaCl, 50 mM Tris (pH 7.5), 0.5% NP-40, protease inhibitors (Sigma), and phosphatase inhibitors (Gene Depot)). Protein concentration was determined using Bio-Rad protein assay reagent. Equal amounts of total protein were loaded in each lane of a 12% SDS-PAGE gel and transferred to polyvinylidene difluoride membranes after electrophoresis. Western blot analyses were performed using the antibodies as indicated. Details and dilutions of antibodies used are described in Supplementary Table 2. Un-cropped scans of key western blot images are available in Supplementary Fig. 8.

**Cycloheximide chase assay.** Cycloheximide chase assay was performed as described previously[37]. In brief, cells were treated with either cycloheximide (25 μg/ml) and DMSO (vehicle control), or Cycloheximide and MCB-613 (6 μM) for the indicated time periods. Cells were harvested at specified time points post-treatment, washed using PBS, and lysed in lysis buffer (150 mM NaCl, 50 mM Tris (pH 7.5), 0.5% NP-40, protease (Sigma), and phosphatase inhibitors (Gene Depot)). Protein concentration was determined using Bio-Rad protein assay reagent. Western blot was performed as described earlier.

**Immunofluorescence analysis.** Ovarian cancer cells cultured on coverslips were fixed in 4% paraformaldehyde for 15 min at room temperature, and permeabilized in 0.1% Triton X-100 in phosphate-buffered saline for 15 min on ice. The coverslips were blocked with 1% BSA, and then incubated overnight at 4 °C in rabbit anti-p53 antibody (1:100) or anti-mouse Calnexin antibody (1:300), followed by incubation with either a FITC or TRITC-conjugated goat anti-rabbit 2° antibody (Vector Labs, 1:1000) for 1 h at room temperature. Hoechst dye was used to stain the nucleus. For lysosome and mitochondria co-localization studies, the cells in culture were treated with either lyotracker or mitotracker dye for 40 and 20 min, respectively. The cells were fixed with 4% paraformaldehyde, permeabilized with Triton X-100, and incubated with appropriate antibodies (see Supplementary Table 2) as described above. Hoechst dye was used to stain the nucleus. Fluorescence images were captured using a GE Healthcare DeltaVision live high-resolution deconvolution microscope.

**Co-immunoprecipitation.** Cells were harvested, washed twice in PBS, and lysed in buffer containing 150 mM NaCl, 50 mM Tris (pH 7.5), 0.5% NP-40, protease, and phosphatase inhibitors. Protein concentration was determined using Bio-Rad protein assay reagents (Bio-Rad Laboratories). Five-hundered microgram of total protein were diluted to a total volume of 500 μl (1 μg/μl) with lysis buffer. Five percent input was removed. The lysate was then pre-cleared using IgG for 2 h at 4 °C followed by 20 μL Protein G sepharose beads for 1 h at 4 °C. Protein G-IgG was pelleted by centrifugation (3000 g; 2 min; 4 °C). Supernatant was transferred to a fresh tube to which anti-p53 antibody (FL-393, 1:100) was added and incubated on a rotator (o/n; 4 °C). Following incubation, 20 μL Protein G sepharose beads were added (1 h, 4 °C). The beads were washed four times with lysis buffer. Bound proteins were eluted by heating at 95 °C for 5 min in 50 μL SDS loading buffer. Immunoblotting was done as described above, using anti-ubiquitin antibody.

**RNAseq and connectivity MAP analysis.** RNA was extracted from MCF-7 cells treated with MCB-613 for 6 h. Paired-end libraries were generated and sequenced using the Illumina HiSeq 2500 Sequencer. Fastq files from the RNAseq data were imported into CLC Genomics Workbench 10 (CLC Bio, Aarhus, Denmark; now Qiagen), quality-controlled, and processed using RNAseq workflow. Sequencing

reads were mapped to Human reference genome GRCh38.p7 and counted. RNA-Seq results from MCF-7 cells, treated with MCB-613 for 6 h, were analyzed using C-MAP to discover perturbations associated with MCB-613 treatment. The DUB inhibitor NSC632839 was the top hit, and has considerably similar chemical structure to MCB-613. The statistics for the RNAseq data is shown in supplementary data, read count statistics (Supplementary Table 4), and fragment counting statistics (Supplementary Table 5). TPM and RPKM were used as the expression values. To test differential expression due to treatment with NSC632839 or MCB-613, the expression level of each gene was estimated by a separate Generalized Linear Model (GLM), and GLM dispersion for a gene was adjusted using the multi-factorial EdgeR method[38]. The fold changes were calculated from the GLM, which was corrected for differences in library size between samples. The Wald Chi-Squared Test was used to find out the statistical significance. Venn diagram (Fig. 3j) was generated using genes with a minimal 3-fold change, and a Bonferroni adjusted maximum p-value of 0.05.

**In vivo ubiquitination assay**. TYK-Nu cells were treated with either 6 μM MCB-613/DMSO for 20 min or with USP15/non-targeting siRNA for 48 h, harvested, washed twice in PBS and lysed in buffer containing 150 mM NaCl, 50 mM Tris (pH 7.5), 0.5% NP-40, protease and phosphatase inhibitors, 5 mM N-ethylmaleimide, and 20 μM MG132. p53 IP is preformed as described above. Immunoblotting was done as described above, using anti-ubiquitin antibody to determine effect of treatment on p53-ubiquitination. For in vivo ubiquitination analysis by His-ubiquitin pulldown, cells were transfected with either His-ubiquitin expression vector alone (TYK-Nu cells) or co-transfected with the His-ubiquitin expression vector, and p53-R175H, expressing plasmid (SKOV3 cells) as described above. Forty-eight hour after transfection, the cells were treated with 6 μM MCB-613 or vehicle (DMSO) for 20 min, harvested, washed twice in PBS and directly lysed in the 8 M urea lysis buffer (containing 20 mM imidazole and benzonase). Cell lysate was cleared by centrifugation and equal amounts of total protein were incubated with 30 μL nickel resin for 1 h at room temperature. After extensive washes using the urea lysis buffer, bound proteins were analyzed by boiling the resin in SDS-PAGE loading buffer containing 250 mM imidazole, followed by SDS-PAGE and immunoblotting, using anti-p53 antibody.

**Fluorescence spectroscopy**. Fluorescence spectroscopy was used to determine whether MCB-613 was bound to p53-R175H mutant protein. Fluorescence experiments were carried out on an ISS PC1 photon-counting spectrofluorometer (ISS, Chicago, IL) with a Peltier automated temperature controller. Intrinsic protein fluorescence measurements were collected using an excitation wavelengths 275 nm and emission wavelengths from 289 to 450 nm. Fluorescence readings were corrected for buffer (PBS) and MCB-613 fluorescence. Recombinant 6xHis-p53-R175H was expressed and purified from bacteria. The purified protein was dialyzed into buffer lacking imidazole. One micrometer recombinant p53-R175H protein and 6 μM MCB-613 were used for the binding assay. The assay was carried out at 25.0 °C ± 0.05.

**Cell viability analysis**. A total of $10^4$ cells were plated in each well of a 96-well plate, and treated with different concentration of specific drugs or siRNA or vehicle controls as indicated. Cell viability was measured at indicated time points using WST1 reagent (Roche), as per manufacturer's instructions. The values reported are mean ± SE of multiple experiments.

**Colony forming assay**. TYK-Nu cells were transfected with either control siRNA or USP15 siRNA, as described earlier. Twenty-four-hour post siRNA transfection, the cells were trypsinized and 2000 cells from each treatment was mixed with Methocult™ colony forming assay media, as per the manufacturer's instructions (Stem cell technologies), and plated onto non-tissue culture treated 6-well plates. The colonies were analyzed under a microscope to determine the number and size of colonies 14 days post seeding. These experiments were performed in quadruplicates.

**DUB activity assay**. Ub-AMC assay was performed to determine the ability of PR-619 and MCB-613 to inhibit USP15 activity. Ub-AMC was purchased from Boston Biochem. Ub-AMC hydrolysis assays were performed in assay buffer (50 mM Tris·HCl, 500 μM EDTA, 5 mM DTT, 0.1% BSA) at 30 °C. Measurements were performed in triplicate with 2.5 nM of recombinant USP15 and 400 nM of substrate (Ub-AMC) in a total volume of 100 μl per well in a 96-well plate. The ability of increasing concentrations to PR-619 and MCB-613 to inhibit USP15 activity was determined by taking measurements, using a Spectramax M2 plate reader (Molecular Devices) with a 360/460 nm filter pair every 40 s for 1 h.

**Data availability**. The authors declare that all the data supporting the findings of this study are available within the article and its supplementary information file and is available from the corresponding author upon request. All RNAseq data presented in this manuscript is available from the GEO repository: GSE110220.

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

## Acknowledgements

This work was supported by NIH grant NIH-CA-181808 to J.S.R. Imaging was supported by the Integrated Microscopy Core at BCM with funding from NIH (DK56338, and CA125123), CPRIT (RP150578), the Dan L. Duncan Comprehensive Cancer Center, and the John S. Dunn Gulf Coast Consortium for Chemical Genomics.

## Author contributions

A.P. and J.S.R. conceived the ideas and designed the experiments. A.P. performed all the experiments. N.C. performed p53 siRNA knockdown. K.-K.W. performed RNAseq and c-Map data analyses. B.C.N., D.M.L., and B.W.O. provided reagents, RNAseq data and many helpful discussions. A.P. and J.S.R. analyzed the results and wrote the manuscript.

## Additional information

**Competing interests:** D.M.L. and B.W.O. are co-founders and hold stock in Coactigon, Inc. which is developing steroid receptor coactivator modulators for clinical use. The remaining authors declare no competing interests.

