## [Peer Review File(PDF 247 kb) · Nature Communications]

Reviewers' comments:

Reviewer #1 (Remarks to the Author):

In this study, Achuth and co-authors showed that treatment with small molecule MCB-613 induces specific depletion of the p53-R175H mutant, but not wt p53 or other tested mutant p53, in ovarian cancer cells by mediating its nuclear export and lysosome-dependent turnover. They further showed that two DUB inhibitors, NSC632839 and PR-619, mimic the effect of MCB-613 on p53-R175H. Using siRNA-mediated knockdown of individual targets of the Dub inhibitors, they identified USP15, but not USP7, is involved in regulating this lysosome-dependent p53-R175R degradation. They verified that USP15 interacts with p53-R175H and knockdown of USP15 results in the increased levels of p53-R175H ubiquitination. Consistently, they showed that USP15 is overexpressed in a number of human cancers through data mining.

Overall, this study is very interesting. The results suggest a novel and specific MCB-613 treatment-induced and USP15-regulated degradation pathway for p53-R175H, thus providing a potential new therapeutics for p53-R175H containing cancers. My concerns are listed below:

1. Lysosome-mediated degradation of p53-R175H is evident and USP15 is implied to suppress this degradation. However, how USP15 regulates p53-R175H ubiquitination, nuclear export and lysosome degradation is not explored.
2. Whether p53-R175H is a direct target of MCB-613 is not known. Does p53-R175H binds to p53? If not, what is the mechanism underlying the MCB-613-mediated p53-R175H degradation? Does MCB-613 target USP15? A clear demonstration of the mechanism underlying the MCB-613's action is critical.
3. Fig.2I: A positive control is needed for MG132 treatment to show that the MG132 treatment indeed worked. In Fig.2J, Lysosome inhibitor treatment partially rescued MCB-613-mediated p53-R175H degradation. Does it imply there are other degradation pathways involved? Does lysosome inhibitor treatment accumulate p53-R175H in lysosome following MCB-613 treatment? Fig. 2B SRC1 knockdown is not convincing.
4. Fig. 3A: Individual channels need to be shown. It seems that the levels of nuclear p53-R175H at 40 min after treatment is higher than the control and are not apparently reduced at 120 min after treatment, which is completely different from results from Fig. 1A.
5. Fig.4 B: input indicates that MCB-613 treatment increase the total levels of ubiquitination in cells, why? Does MCB-613 also increase the ubiquitination of p53 mutants other than p53-R175H?
6. Fig.5: The authors showed the structural similarity between MCB-613 and NSC632839, since PR-619 has similar effect, does it imply that MCB-613 is actually a Dub inhibitor?
7. Fig.6: The authors showed that USP15 KD induces the degradation of p53-R175H, but not R273H. How about p53-R248Q in OVCAR3 cells, given that the authors used OVCAR3 in cell viability assays (Fig. 6M)?
8. Fig. 6F: Co-IP between USP36 and p53-R273H is not convincing, normal IgG control is needed. The interaction of USP15 with other p53 mutants, including p53-R273H and p53-R248Q, should be similarly examined. Does USP15 also interact with wt p53?
9. Fig. 6G: There is no p53 expression shown in input and p53 IP. The expression of USP15 does not make sense. Fig.6H is not well-presented, no expression data for USP15, p53 and loading control was shown.

10. Fig. S4D: The authors indicated “none of the single lysine substitution mutants we tested resulted in resistance to MCB-613 induced turnover of the R175H mutant”. However, R175H/K370R and R175H/K381R seems to be partially resistant to MCB-613 treatment. It’s worthy to check ubiquitination of these mutants upon MCB-613 treatment.

11. Fig. 7A revealed USP15 expression is significantly elevated in several types of cancers. Does this increase correlate with the p53-R175H mutant? What’s the frequency of the p53-R175H mutant in these cancers?

Reviewer #2 (Remarks to the Author):

In this manuscript, the authors investigated the effect of MCB-613 on the protein stability of gain-of-function (GOF) p53-R175H mutant. They demonstrated that MCB-613 treatment decreases the protein abundance of GOF p53-R175H mutant through lysosomal degradation. The authors found that MCB-613 has an antagonistic effect of USP15; thus, they predicted that USP15 dependent deubiquitination is the mechanism of p53-R175H stabilization and aberrant activation. The data showing the selective downregulation of p53-R175H with MCB-613 is convincing. However, the physiological relevance of USP15 on GOF p53-dependent cancer progression is not well defined. There are several comments that should be addressed.

1. What is the molecular mechanism underlying the p53-WT protein stabilization upon MCB-613 treatment in Figure 2H? If this is due to concurrent Mdm2 down regulation, the author should include a WB blot panel of p53 in Figure 4K to indicate that p53-WT protein accumulates in a time-dependent manner upon MCB-613 treatment.
2. Figure 4: The author should demonstrate which type of ubiquitin linkage the p53-R175H is subjected. As they claim that the degradation is through the lysosomal pathway, it is predicted that K63-linkage poly-ubiquitination is more dominant than that of K48-linkage.
3. Figure 6: Multiple USP15 siRNAs should be included in the assays for excluding the possibility of off-target effects.
4. Figure 6M: Analysis of only cell viability is not sufficient to claim the oncogenic function of the USP15/GOF p53 signaling that contributes to ovarian cancer development and progression. The authors should conduct several critical biological experiments such as anchorage-dependent and independent cell proliferation assays and xenograft tumorigenesis assay, and/or cell migration and invasion assays, with GOF p53-expressing cell lines in which USP15 is stably depleted by shRNA or CRISPR/Cas.
5. It would be helpful if each figure panel provides p53 gene status of ovarian cancer cell lines, as is done in Figure 2G and 2H.

Reviewers' comments:

Reviewer #1:

In this study, Achuth and co-authors showed that treatment with small molecule MCB-613 induces specific depletion of the p53-R175H mutant, but not WT p53 or other tested mutant p53, in ovarian cancer cells by mediating its nuclear export and lysosome-dependent turnover. They further showed that two DUB inhibitors, NSC632839 and PR-619, mimic the effect of MCB-613 on p53-R175H. Using siRNA-mediated knockdown of individual targets of the DUB inhibitors, they identified USP15, but not USP7, is involved in regulating this lysosome-dependent p53-R175R degradation. They verified that USP15 interacts with p53-R175H and knockdown of USP15 results in the increased levels of p53-R175H ubiquitination. Consistently, they showed that USP15 is overexpressed in a number of human cancers through data mining.

Overall, this study is very interesting. The results suggest a novel and specific MCB-613 treatment-induced and USP15-regulated degradation pathway for p53-R175H, thus providing a potential new therapeutics for p53-R175H containing cancers.

Response: We thank the Reviewer for his/her many thoughtful comments and suggestions.

Specific comments:

1. *Lysosome-mediated degradation of p53-R175H is evident and USP15 is implied to suppress this degradation. However, how USP15 regulates p53-R175H ubiquitination, nuclear export and lysosome degradation is not explored.*

Response: As pointed out by Reviewer 1, the data presented in our manuscript provide definitive evidence to show that the small molecule MCB-613 causes a selective increase in p53-R175H ubiquitination and targets the ubiquitinated protein for lysosome-mediated degradation. Further, we show that the effect of MCB-613 on p53-R175H is mediated through deubiquitinase (DUB) activity and identify USP15 as a p53-R175H specific DUB. We show that USP15 knockdown causes increased ubiquitination of p53-R175H GOF mutant but not p53-R273H (Fig. 6H and 6I). PR-619, a small molecule capable of inhibiting USP15 activity (new data, Fig. S6C) causes nuclear export of p53-R175H and a decrease in p53-R175H levels (Fig. 5F and 5I). USP15 knockdown causes a decrease in p53-R175H levels (Figs. 6D, S6A and 6J), which is reversed upon co-treatment with lysosome inhibitors showing p53-R175H is targeted to the lysosome upon USP15 knockdown (Fig. 6J). In new data that we have added, we also show that the small molecule MCB-613 causes p53-R175H turnover by reducing USP15 protein levels (Fig. 6K).

To address the question as to how precisely USP15 is impacting p53-R175H stability, we feel that it is important to know which sites on p53-R175H are responsible for mediating this effect. To address this question, we used N-terminal and C-terminal deletion constructs of p53-R175H and identified that either deletion of N-terminal 41 amino acids or C-terminal 30 amino acids enables the truncated p53-R175H mutant to resist MCB-613 induced turnover (Fig. S4D). These regions could be important by either serving as ubiquitination sites (lysines in this region) or by mediating protein-protein interactions. Since, the C-terminal lysines in WT-p53 are known to be ubiquitinated and important for mediating protein stability, we made specific lysine to arginine substitution mutants of p53-R175H (S4E and S4F). However, none of the single lysine substitution mutants that we generated resisted the effect of MCB-613 on p53-R175H turnover (Fig. S4F). Since redundancies in ubiquitination sites are known in WT-p53, we made p53-R175H mutants in which multiple lysines were simultaneously mutated to arginine. However, all the mutants we have so far tested failed to resist MCB-613 induced turnover of p53-R175H. This may be due to (1) redundancy in lysine residues in p53-R175H, or (2) the effect of USP15 (and MCB-613 and the DUB inhibitors) could be mediated by cumulative effect on multiple lysines. p53 has 22 potential ubiquitination sites (21 lysines and N-terminus). This would give rise to a large number of possibilities that might need to be evaluated (i.e 22

single lysine mutants, 231 double lysine mutants, 1540 triple lysine mutants and so on). Generating these many mutants and testing them systematically is technically challenging and beyond the scope of this study.

We are currently pursuing/interested in pursuing some of key questions such as, what are the ubiquitin ligases upstream of the p53 mutants that may be involved in the selective turnover of specific mutants? What is the biological consequence of localization of p53-R175H at the endoplasmic reticulum on its way to the lysosome? Are there chaperone proteins involved in this process? If so, what are they? However, we feel that addressing these questions carefully is a manuscript or more in itself and thus beyond the scope of this paper. We believe that the findings presented in this manuscript highlight an important aspect of mutant p53 regulation and can have a potentially huge impact on further analysis of the different mutant p53 proteins. It is our hope that the present study will open up more provocative and important questions in the field such as the ones Reviewer 1 has proposed and pave way for more mechanistic studies on the different GOF mutant p53 proteins.

2. Whether p53-R175H is a direct target of MCB-613 is not known. Does p53-R175H binds to p53? If not, what is the mechanism underlying the MCB-613-mediated p53-R175H degradation? Does MCB-613 target USP15? A clear demonstration of the mechanism underlying the MCB-613's action is critical.

Response: The questions raised by Reviewer 1 are definitely important and prompted us to further characterize the mechanism through which MCB-613 is mediating its effect on p53-R175H. Our additional experiments now show that MCB-613 impacts p53-R175H levels by causing a decrease in USP15 protein levels (but not mRNA). The details of the specific experiments we performed to address Reviewer 1's specific questions are explained below. The data generated from these experiments are now integrated in the manuscript as indicated.

- (i) p53-R175H does not bind to MCB-613: To test whether p53-R175H is a direct target of MCB-613 and to determine if p53-R175H binds to MCB-613, we cloned p53-R175H with a N-terminal 6xHis tag in the bacterial expression vector pQE80L, which allowed us to express the p53-R175H protein in bacteria. The recombinant 6xHis-p53-R175H protein was purified by Ni-NTA affinity chromatography. The purified protein was then dialyzed to a buffer that lacks imidazole. The binding of the dialyzed recombinant p53-R175H to MCB-613 was tested using a fluorescence binding assay. Our results reveal that MCB-613 does not bind to p53-R175H. These results are now included in the manuscript as Fig. S4H.
- (ii) This result is consistent with and further supports our *in vivo* results. We had previously tested the possibility that MCB-613 binds to p53-R175H and causes 'reactivation' of p53-R175H (binds and causes conformational changes in p53-R175H to make it more WT-p53 like). If MCB-613 causes reactivation of p53-R175H, we would expect MCB-613 to cause decrease in p53-R175H levels through the MDM2 mediated pathway. Our results using MDM2 siRNA (Figs. 4I, 4J, and S4I) and nutlin-3A (Fig. S4G) revealed that this is not the case.
- (iii) MCB-613 does not alter USP15 catalytic activity: Since MCB-613 does not bind p53-R175H, as per Reviewer 1's suggestion, we sought to identify how MCB-613 impacts USP15 function. First, we tested the possibility that MCB-613 acted as a direct inhibitor of USP15. To test this possibility, we expressed 6xHis tagged USP15 in bacteria and purified the protein by Ni-NTA affinity chromatography. The purified USP15 was used in *in vitro* deubiquitination assay with Ub-AMC substrate. Using this assay we tested the ability of MCB-613 to inhibit the activity of USP15. PR-619, a known pan-DUB inhibitor that is also known to inhibit USP15 served as a positive control. Our results show that MCB-613 does not inhibit USP15 activity. These data are now included in the manuscript as Fig. S6C and S6D.
- (iv) MCB-613 does not alter USP15 mRNA levels: We then tested whether MCB-613 causes a decrease in USP15 transcript levels. Cells expressing p53-WT (ALST), p53-R175H (TYK-Nu), p53-R248Q (OVCA3), p53-R273H (OVCA420) were treated with MCB-613 for 2h. RNA was

extracted using Qiagen RNAeasy kit as per the manufacturer's protocol and cDNA was synthesized. Quantitative qRT-PCR analyses using cDNA from control (DMSO) and MCB-613 treated samples revealed that MCB-613 treatment had no effect on USP15 mRNA levels (Fig. S6E).

- (v) MCB-613 causes decrease in USP15 protein levels: To determine if MCB-613 altered USP15 protein levels in TYK-Nu cells (p53-R175H), we treated ovarian cancer cells with MCB-613 for 2 hours. Western blot analysis of cell lysates showed that MCB-613 causes a decrease in USP15 levels in TYK-Nu (p53-R175H) and OVCA420 (p53-R273H) cells. These results are now included in the manuscript as Fig. 6K.

3. Fig.2I: A positive control is needed for MG132 treatment to show that the MG132 treatment indeed worked.

Response: As per Reviewer 1's suggestion, we repeated the experiment and have now included an anti-ubiquitin Western blot to show that the MG132 treatment worked (Fig. 2I). Both MCB-613 and MG132 treatment resulted in increased accumulation of poly-ubiquitinated proteins. MG132, as previously described, did not rescue MCB-613 induced degradation of p53-R175H.

4. In Fig.2J, Lysosome inhibitor treatment partially rescued MCB-613-mediated p53-R175H degradation. Does it imply there are other degradation pathways involved?

Response: The ability of lysosomal inhibitors to impair the loss of p53-R175H protein is consistent and reproducible. In response to reviewer's suggestion, we repeated the experiment again and found the rescue to be very strong (Fig. 2J). But as suggested by the reviewer, the effect may be slightly less than 100%. It could be potentially due to (1) the effectiveness of these inhibitors and (2) the kinetics of lysosome inhibitors compared to the effect of MCB-613 on p53-R175H (which is very rapid) and (3), in the absence of any evidence against it, we also cannot rule out possibility of other potential mechanisms. We have now included these points in our discussion section to reflect the different possibilities (Under 'Discussion section'; para 4, line 13). We thank Reviewer 1 for bringing this up.

5. Fig. 2B SRC1 knockdown is not convincing.

Response: Different cell lines express different levels of the steroid receptor co-activators: SRC1, SRC2 and SRC3. The expression level of SRC1 is very low (if any) in ALST cells and hence the effect of SRC1 knock down is not detected on the Western blot in Fig. 2B. We regret not making this point clear in the text of our manuscript and we apologize for not doing so. The text is revised to explain why SRC1 knockdown in ALST cells in Fig 2B appears the way it does (Pg 2, para 3, line 13). Thank you for pointing this out to us.

6. Fig. 3A: Individual channels need to be shown. It seems that the levels of nuclear p53-R175H at 40 min after treatment is higher than the control and are not apparently reduced at 120 min after treatment, which is completely different from results from Fig. 1A.

Response: As per reviewer 1's suggestion, the individual channels are now shown for Fig. 3A and Fig.3B in (Fig. S3A and S3B).

Fig.1A is representative of the total protein in the population of cells being analyzed (Western blot). The nuclear export (and subsequent turnover) of p53-R175H appears at different times in different cells, probably reflective of the pharmacokinetics of the drug. In some cells we see a loss early as 10 minutes, while in others we see a loss at later time points (40 min or 60 min). In this figure we focused on cells that still retain p53 protein at the indicated time-points to show differences in their localization. A wide field (low magnification) reflective of the Western blot in Fig. 1A is shown in Fig 1I and clearly shows a decrease in p53-R175H upon 120 min MCB-613 treatment. In Fig. 3A we focused on cells that still retain some p53-R175H (even at 120 min) to show the difference in p53-R175H localization in response to MCB-613

treatment. As seen in Fig. 1I, some TYK-Nu cells still retain some p53-R175H protein even after 120 min of MCB-613 treatment.

7. Fig.4 B: input indicates that MCB-613 treatment increase the total levels of ubiquitination in cells, why? Does MCB-613 also increase the ubiquitination of p53 mutants other than p53-R175H?

Response: As indicated by Reviewer 1, MCB-613 treatment results in an increase in total levels of ubiquitination in cells. This is consistent with previously published results¹. Wang L et.al, (Cancer Cell, 2015) demonstrate that MCB-613 causes cellular stress including ER-stress and oxidative damage. Such stresses are known to cause the accumulation of ubiquitinated proteins in cells. We speculate that this might be the reason for increased total ubiquitination in cells upon MCB-613 treatment. Thus our result is consistent with previous results.

Upon MCB-613 treatment, we observed a rapid increase in ubiquitination of p53-R175H mutant protein. In contrast, we did not observe increase in ubiquitinated forms of p53-WT or p53-R273H mutant proteins (Fig. 4A).

References:

1. Wang, L. *et al.* Characterization of a Steroid Receptor Coactivator Small Molecule Stimulator that Overstimulates Cancer Cells and Leads to Cell Stress and Death. *Cancer cell* **28**, 240-252 (2015).

8. Fig.5: The authors showed the structural similarity between MCB-613 and NSC632839, since PR-619 has similar effect, does it imply that MCB-613 is actually a Dub inhibitor?

Response: We tested the ability of MCB-613 to inhibit USP15 activity and found that MCB-613 does not act directly to inhibit USP15 (Fig. S6C and S6D). Rather, we show in our revised manuscript that MCB-613 impacts USP15 by causing a rapid decrease in its protein level (Fig. 6K). However, we have not tested the possibility that MCB-613 could act as an inhibitor DUBs other than USP15. Although understanding the ability of MCB-613 to inhibit the activity of specific DUBs would be interesting, we believe that systematically testing the ability MCB-613 to inhibit individual DUBs is beyond the scope of this study.

9. Fig.6: The authors showed that USP15 KD induces the degradation of p53-R175H, but not R273H. How about p53-R248Q in OVCAR3 cells, given that the authors used OVCAR3 in cell viability assays (Fig. 6M)?

Response: We performed USP15 siRNA in OVCAR3 cells (p53-R248Q) to address Reviewer 1's question. Our result (Fig. 6F) shows that USP15 knockdown does not alter p53-R248Q levels in OVCAR3 cells.

10. Fig. 6F: Co-IP between USP15 and p53-R273H is not convincing, normal IgG control is needed. The interaction of USP15 with other p53 mutants, including p53-R273H and p53-R248Q, should be similarly examined. Does USP15 also interact with wt p53?

Response: We apologize for not including the normal IgG controls earlier. We have included all necessary controls in the revised figure. In response to the reviewer 1's question on whether USP15 interacts p53-WT, p53-R248Q and p53-R273H we immunoprecipitated p53 from ALST (p53-WT), OVCAR3 (p53-R248Q) and OVCA420 (p53-R273H) cells and performed a USP15 Western blot to determine if USP15 co-IP'd with WT or the other GOF mutant p53 protein. Our results show that USP15 does not interact with p53-WT or p53-R248Q or p53-R273H. These results are now included as Fig. S6B. SKOV3 (p53-null) served as a control for p53-IP.

11. Fig. 6G: There is no p53 expression shown in input and p53 IP. The expression of USP15 does not make sense. Fig.6H is not well-presented, no expression data for USP15, p53 and loading control was shown.

Response: We have now corrected this omission and revised the figure as per Reviewer 1's suggestion. We have also revised Fig. 6I (previously Fig. 6H) and hope the revised figure addresses the issues pointed

our by Reviewer 1. We have also included USP15, p53 and β -actin loading control as suggested by Reviewer 1.

12. Fig. S4D: The authors indicated “none of the single lysine substitution mutants we tested resulted in resistance to MCB-613 induced turnover of the R175H mutant”. However, R175H/ K370R and R175H/K381R seems to be partially resistant to MCB-613 treatment. It’s worthy to check ubiquitination of these mutants upon MCB-613 treatment.

Response: We repeated these experiments and found that they did not rescue the effect of MCB-613 on the p53-R175H GOF mutant. To further confirm, we made a triple mutant, p53-R175H: K370R: K381R and found the triple mutant was also readily degraded upon MCB-613 treatment. These results are now included in Fig. S4F.

13. Fig. 7A revealed USP15 expression is significantly elevated in several types of cancers. Does this increase correlate with the p53-R175H mutant? What’s the frequency of the p53-R175H mutant in these cancers?

Response: Analyses using data retrieved from COSMIC database (Catalogue Of Somatic Mutations In Cancer - <http://cancer.sanger.ac.uk/cosmic>) release v82 shows that among 1936 TCGA samples with p53 mutations, 83 cases have the R175H GOF mutation, representing 4.29% of all p53 mutation containing human cancers. Among these the frequencies of the different cancers exhibiting increased USP15 levels are as follows: Ovarian cancer (8/358 = 2.23%); Glioblastoma (7/161 = 4.35%); Breast carcinoma (18/297 = 6%) and prostate cancer (2/20 = 10%). We have now included this information in our manuscript. These data confirm that p53-R175H mutation is frequently observed in several human cancers and that an approach that can target cancer cells carrying these mutations can pave the way for personalized therapy and will be clinically important.

Although the above data suggests that USP15 overexpression and p53-R175H mutations could co-exist in these cancers, it is not possible to definitively state using the data available that they co-exist in the same patient. To address whether the p53-R175H mutation correlates with an increase in USP15 expression in the same patients, we attempted to analyze publically available clinical data sets for correlation between p53 mutation and USP15 expression within the same sample. Our ability to perform these analyses was severely limited by the availability of datasets that contain both p53 mutation status and USP15 expression levels from the same sample. We used data available from COSMIC database release v82, which is updated periodically and reflects the current status of data that is available publically. As shown below, our analysis suggests a trend towards the existence of p53-R175H mutations and USP15 expression in ovarian cancer. However, the available sample size is too low to perform reliable statistical analyses (p-value not reliable as sample size of cases with p53-R175H mutation is very low (n=7)). Due to these limitations, we do not feel confident to include or comment on this observation in the current manuscript. It may be appropriate to perform such an analysis in future when more clinical data becomes available. We have summarized the data below for the reviewers.

USP15 expression in ovarian cancer samples with the following p53 status:

p53 status of ovarian cancer specimen	R175H (n=7)	WT (n=96)	All mutations together (n=292)
USP15 expression (TPM)	8.65	7.50	7.67
p-value (compared to R175H)		0.140178	0.189091

TPM – Transcript Per Million is being used as expression value. Data was retrieved from the processed TCGA RNAseq data (GEO database: GSM1536837)

It should also be noted that for USP15 to serve as an effective therapeutic target in p53-R175H expressing tumor, it may not be necessary for USP15 to be overexpressed. Our data show that knockdown (and inhibition) of prevalent/basal USP15 activity in ovarian cancer cells results in an increased turnover of p53-R175H and cancer cell death. Thus, strategies that achieve depletion of basal/prevalent USP15 activity could also potentially have important therapeutic application in p53-R175H mutation containing tumors.

However, if USP15 is simultaneously overexpressed in p53-R175H expressing tumors, this effect could definitely be amplified.

Reviewer #2:

In this manuscript, the authors investigated the effect of MCB-613 on the protein stability of gain-of-function (GOF) p53-R175H mutant. They demonstrated that MCB-613 treatment decreases the protein abundance of GOF p53-R175H mutant through lysosomal degradation. The authors found that MCB-613 has an antagonistic effect of USP15; thus, they predicted that USP15 dependent deubiquitination is the mechanism of p53-R175H stabilization and aberrant activation. The data showing the selective downregulation of p53-R175H with MCB-613 is convincing. However, the physiological relevance of USP15 on GOF p53-dependent cancer progression is not well defined. There are several comments that should be addressed.

We thank Reviewer #2 for the many useful comments. Detailed response to the specific questions and suggestions raised by Reviewer 2's are below:

Specific comments:

1. *What is the molecular mechanism underlying the p53-WT protein stabilization upon MCB-613 treatment in Figure 2H? If this is due to concurrent Mdm2 down regulation, the author should include a WB blot panel of p53 in Figure 4K to indicate that p53-WT protein accumulates in a time-dependent manner upon MCB-613 treatment.*

Response: As per Reviewer 2's suggestion we have now included the corresponding p53 blot showing a concurrent increase in p53-WT protein in a time dependent manner in Figure 4K. We had previously not included this data thinking it would be repetitive. However, we now realize that including the data in Fig.4K as suggested by Reviewer 2 will help the readers infer the results more easily. We have also made corresponding changes in text and figure legends to reflect this change.

2. *Figure 4: The author should demonstrate which type of ubiquitin linkage the p53-R175H is subjected. As they claim that the degradation is through the lysosomal pathway, it is predicted that K63-linkage poly-ubiquitination is more dominant than that of K48-linkage.*

Response: The reviewer's point is well taken. As suggested, we tested whether MCB-613 causes the accumulation of K-48 or K-63 ubiquitin chain linkage. In brief, we co-expressed p53-R175H and either His-tagged K-48 only Ubiquitin (all other K's are mutated to R) or His-tagged K-63 only Ubiquitin (all other K's are mutated to R). The cells were then treated with DMSO (vehicle control) or MCB-613 for 15 minutes. Post-treatment, cells were lysed and total ubiquitinated protein containing the His-tagged ubiquitin was pulled down using Ni-NTA chromatography. p53 western blot was performed on the eluate to determine changes in K-48 and K-63 ubiquitinated p53-R175H protein upon MCB-613 treatment.

Our results show that MCB-613 treatment resulted in an increase in both K-48 and K-63 linked ubiquitin chains on p53-R175H protein. However, as Reviewer 2 suspected, the increase in K-63 linked Ub-chains is more dominant. We have now included these results in the manuscript (Fig. S4B and Fig. S4C). We have also modified the text in the manuscript to reflect this data. We thank Reviewer 2 for the insightful suggestion.

3. *Figure 6: Multiple USP15 siRNAs should be included in the assays for excluding the possibility of off-target effects.*

Response: As per the reviewer's suggestion we used shRNA-targeting USP15 to knockdown USP15 and determined its effect on p53-R175H levels. Consistent with our previous results, USP15 knockdown using multiple shRNAs resulted in decreased p53-R175H levels. These results have now been added to the manuscript as Fig. S6A.

4. *Figure 6M: Analysis of only cell viability is not sufficient to claim the oncogenic function of the USP15/GOF p53 signaling that contributes to ovarian cancer development and progression. The authors*

should conduct several critical biological experiments such as anchorage-dependent and independent cell proliferation assays and xenograft tumorigenesis assay, and/or cell migration and invasion assays, with GOF p53-expressing cell lines in which USP15 is stably depleted by shRNA or CRISPR/Cas.

Response: In response to Reviewer 2's suggestion, we performed colony-forming assays to test the effect of USP15 knockdown on the ability of p53-R175H expressing cells to form colonies in soft agar. As our results show, USP15 knockdown significantly reduced the colony forming ability of TYK-Nu (R175H) cells (Fig. 7E). Further, USP15 knockdown also caused visible reduction in the colony size. These results are consistent with our hypothesis and results obtained using cell viability assay. We have now included these results in the manuscript and have also made the required changes in the text to reflect the new data. Although we agree with the reviewer that extending these results to animal models would be very insightful, xenograft experiments have caveats and are beyond the scope of the current manuscript. We believe the results presented are very convincing and establish USP15 as an important and selective upstream regulator of the p53-R175H mutant protein. Given that no specific selective upstream regulator of the different clinically important p53 GOF mutants are known so far, our report provides an important first step towards understanding how the different mutants are regulated in cells and will hopefully pave way to the discovery of new strategies to target them therapeutically.

5. It would be helpful if each figure panel provides p53 gene status of ovarian cancer cell lines, as is done in Figure 2G and 2H.

Response: We agree with the reviewer that clearly indicating the specific mutation status of ovarian cancer cell lines in each figure panel will make it easier for the reader to follow the results. As suggested by Reviewer 2, we have now included this information in our figure panels.

Reviewers' comments:

Reviewer #1 (Remarks to the Author):

I appreciate the authors' efforts to address most of the reviewers' concerns experimentally. However, the mechanism by which MCB-623 degrades p53K175H through ubiquitination-mediated lysosome degradation is still not clear. The authors made effort to demonstrate that MCB-623 does not binds to p53K175H and does not inhibit USP15's Dub activity, but reduces the levels of USP15 protein (Fig. 6K). The authors should at least explore the lysosome degradation pathways such as autophagy. Fig. 6K is important but the result seems not to be convincing and raises additional questions. Why USP15 reduction by MCB-623 in OVCA420 (p53R273H) cells does not reduce p53 levels? Why MCB-623 does not reduces USP15 in other cell lines? How many biological replicate experiments have been done? As the decrease of USP15 is subtle, quantitative assessment of the reduction is critical, particularly this is so far the only potential mechanism explaining the finding. What about the half-life of USP15 upon MCB-623 treatment? How does MCB-623 cause USP15 reduction?

Minor concerns:

New Fig. 4K: Why MG132 treatment did not induce wt p53 levels?

Fig. 6G 6H 6I: Are all the control IgG from the same experiments as anti-p53 IP? If not, the author should preform the assays with anti-p53 antibody and IgG control together.

The last Fig. S6D should be Fig. S6E.

Reviewer #2 (Remarks to the Author):

The authors have addressed the questions in a satisfactory manner. The manuscript is suitable for publication.

Reviewer #1 (Remarks to the Author):

I appreciate the authors' efforts to address most of the reviewers' concerns experimentally. However, the mechanism by which MCB-623 degrades p53K175H through ubiquitination-mediated lysosome degradation is still not clear. The authors made effort to demonstrate that MCB-613 does not binds to p53R175H and does not inhibit USP15's Dub activity, but reduces the levels of USP15 protein (Fig. 6K).

Response: We thank the Reviewer for his/her many thoughtful comments and suggestions.

Specific comments:

1. The authors should at least explore the lysosome degradation pathways such as autophagy.

Response: In response to Reviewer 1's comment we tested the ability of the autophagy inhibitor LY294002 to rescue the effect of MCB-613 on p53-R175H in TYK-Nu cells. Our results show that the autophagy inhibitor LY294002 rescues MCB-613 induced turnover of p53-R175H. This result has been now included in the revised manuscript as Fig. S2E. Corresponding changes in the text have also been made in the text (page 5, under sub-heading- MCB-613 causes nuclear export and lysosome mediated turnover of p53-175H; lines 159-162).

It is interesting to note that a previous study also showed that MCB-613 induced the formation of autophagosomes in breast cancer cells further supporting our new data¹.

References:

1. Wang, L. *et al.* Characterization of a Steroid Receptor Coactivator Small Molecule Stimulator that Overstimulates Cancer Cells and Leads to Cell Stress and Death. *Cancer cell* **28**, 240-252 (2015).

2. Fig. 6K is important but the result seems not to be convincing and raises additional questions. Why USP15 reduction by MCB-613 in OVCA420 (p53R273H) cells does not reduce p53 levels?

Response: As pointed out by Reviewer 1, MCB-613 causes a decrease in USP15 levels in both TYK-Nu (R175H) and OVCA420 cells (R273H). However, only the p53-R175H levels decreases upon MCB-613 treatment. MCB-613 has no effect on p53-R273H levels. These results are expected and consistent with our finding that USP15 selectively regulates of p53-R175H. Some of the key results illustrating this finding are:

- (i) USP15 knockdown causes a decrease in p53-R175H levels but not p53-R273H (Figs. 6D and 6E).
- (ii) USP15 knockdown causes increased ubiquitination of p53-R175H protein but not p53-R273H (Figs. 6H and 6I).

Collectively, these results confirm that USP15 is a selective upstream regulator of p53-R175H levels but not p53-R273H (or p53-R248Q). Thus, although MCB-613 causes a decrease in USP15 levels in both cell lines, we only see (and would expect to see) the p53 (p53-R175H) levels decrease in the TYK-Nu cells.

3. How many biological replicate experiments have been done? As the decrease of USP15 is subtle, quantitative assessment of the reduction is critical, particularly this is so far the only potential mechanism explaining the finding.

Response: The above experiment has been repeated three times. As per Reviewer 1's suggestion the Western blots were quantified using image J and the normalized Western blot quantifications are now included as Fig. S6F.

4. What about the half-life of USP15 upon MCB-613 treatment? How does MCB-613 cause USP15 reduction?

Response: The questions raised by reviewer 1 are extremely interesting and we are currently pursuing/interested in pursuing some of these questions. Given our discovery that USP15 is an important regulator of the clinically important gain-of-function p53-R175H mutant, understanding how USP15 is regulated is extremely important. To our knowledge there are no previous reports on regulation of USP15 protein. However, we feel that addressing these questions carefully is a manuscript or more in itself and thus far beyond the scope of this paper. We believe that the findings presented in this manuscript highlight an important aspect of mutant p53 regulation and identifies USP15 as a key upstream regulator of p53-R175H. It is our hope that the results presented in the current manuscript will encourage other researchers (in addition to us) to divert their efforts towards understanding USP15 regulation in cells.

Minor concerns:

5. *New Fig. 4K: Why MG132 treatment did not induce wt p53 levels?*

Response: MCB-613 causes a drastic decrease in the levels of MDM2, an E3 ligase that has been shown to mediate ubiquitination and turnover of WT-p53. As expected the p53-WT levels increased upon MCB-613 treatment. However, upon co-treatment of MG132 (proteasome inhibitor) with MCB-613, the MCB-613 induced depletion of MDM2 was reversed suggesting MDM2 is targeted for proteasomal degradation upon MCB-613 treatment. Rescuing MDM2 degradation using MG132 should restore p53-WT degradation pathway but p53-WT levels still remain higher because MG132 inhibits the proteasome and hence MDM2-mediated proteasomal degradation of p53-WT is also blocked.

6. *Fig. 6G 6H 6I: Are all the control IgG from the same experiments as anti-p53 IP? If not, the author should perform the assays with anti-p53 antibody and IgG control together.*

Response: All the IgG controls are from the same experiments as anti-p53 IP.

7. *The last Fig. S6D should be Fig. S6E.*

Response: We apologize for the error and thank Reviewer 1 for pointing this out to us. We have corrected this error in the revised manuscript.

Reviewer #2 (Remarks to the Author):

The authors have addressed the questions in a satisfactory manner. The manuscript is suitable for publication.

Response: We are happy to know that the reviewer found our responses satisfactory. We thank the Reviewer for his/her many thoughtful comments and suggestions.

REVIEWERS' COMMENTS:

Reviewer #1 (Remarks to the Author):

The authors have properly addressed my concerns and the paper is now acceptable.